# Considerations, Advances, and Challenges Associated with the Use of Specific Emitter Identification in the Security of Internet of Things Deployments: A Survey

**Joshua H. Tyler** [†][iD]**, Mohamed K. M. Fadul** [†] **and Donald R. Reising** *,[†][iD]

Electrical Engineering Department, College of Engineering & Computer Science, The University of Tennessee at Chattanooga, Chattanooga, TN 37403, USA; ygm111@mocs.utc.edu (J.H.T.); mohammed-fadul@utc.edu (M.K.M.F.)
* Correspondence: donald-reising@utc.edu
† These authors contributed equally to this work.

**Abstract:** Initially introduced almost thirty years ago for the express purpose of providing electronic warfare systems the capabilities to detect, characterize, and identify radar emitters, Specific Emitter Identification (SEI) has recently received a lot of attention within the research community as a physical layer technique for securing Internet of Things (IoT) deployments. This attention is largely due to SEI's demonstrated success in passively and uniquely identifying wireless emitters using traditional machine learning and the success of Deep Learning (DL) within the natural language processing and computer vision areas. SEI exploits distinct and unintentional features present within an emitter's transmitted signals. These distinctive and unintentional features are attributed to slight manufacturing and assembly variations within and between the components, sub-systems, and systems comprising an emitter's Radio Frequency (RF) front end. Although sufficient to facilitate SEI, these features do not hinder normal operations such as detection, channel estimation, timing, and demodulation. However, despite the plethora of SEI publications, it has remained largely a focus of academic endeavors, primarily focusing on proof-of-concept demonstration and little to no use in operational networks for various reasons. The focus of this survey is a review of SEI publications from the perspective of its use as a practical, effective, and usable IoT security mechanism; thus, we use IoT requirements and constraints (e.g., wireless standard, nature of their deployment) as a lens through which each reviewed paper is analyzed. Previous surveys have not taken such an approach and have only used IoT as motivation, a setting, or a context. In this survey, we consider operating conditions, SEI threats, SEI at scale, publicly available data sets, and SEI considerations that are dictated by the fact that it is to be employed by IoT devices or IoT infrastructure.

**Keywords:** specific emitter identification; radio frequency fingerprinting; physical layer authentication; physical layer security; Internet of Things

## 1. Introduction

The Internet of Things (IoT) consists of two key components: (i) semi-autonomous devices that leverage inexpensive computing, networking, sensing, and actuating capabilities to sense and carry out actions within the physical world and (ii) connection to the Internet [1]. It is important to note that our use of "IoT" encompasses the Internet of Battlefield Things (IoBT), Internet of Military Things (IoMT), Industrial IoT (IIoT), Internet of Vehicles (IoV), and other devices that satisfy the above definition. By 2025, the number of deployed IoT devices is projected to reach seventy-five billion [2–4]. Continued IoT deployments create an even larger surface over which bad actors can conduct attacks to carry out nefarious activities and exploit individuals or sets of IoT devices and their associated infrastructure. Disturbingly, most IoT devices employ weak or no encryption [5]. The use of weak or no encryption is attributed to (i) limited on-board computational resources (e.g.,

---



memory, power, etc.), (ii) prohibitive manufacturing costs, and (iii) scalability challenges associated with implementation and key management [5–7]. Weak or lack of encryption is being successfully exploited and abused [8–15]. In light of this information, there is a critical need for an effective way to secure IoT devices and their corresponding infrastructure. One solution is a physical layer-based approach known as Specific Emitter Identification (SEI) [16–18].

Almost thirty years ago, SEI was introduced to provide electronic warfare systems the functionality of detecting, characterizing, and identifying radar systems via immutable features present within their transmitted signals [19–23]. These immutable features have been attributed to the components, sub-systems, and systems comprising the radar's Radio Frequency (RF) front end. Figure 1 provides a representative illustration of the specific signal features that an SEI process can exploit. Based upon this visualization and the nature of the features' origins, the collection, generation, grouping, or learning of these specific features are commonly referred to as an RF fingerprint or RF-Distinct, Native, Attributes (RF-DNA) fingerprint. RF-DNA captures the essence of an emitter's identity in much the same way a person's DNA is essential to determining who an individual is in terms of traits, features, and so on. SEI is advantageous due to its (i) passive nature, which means that the targeted emitter generates signals, as part of its intended mission, without external stimulation, (ii) exploitation of distinct, unique, and organic features that are unintentionally imparted to the transmitted signal by the target emitter's RF front-end components, (iii) ability to measure the exploited features present within the signal quantitatively, and (iv) exploitation of persistent features across time, location, and environments. The success of radar SEI led to it being adopted as a potential means to augment higher-level (e.g., encryption, MAC address filtering, etc.) security mechanisms employed within private and public wireless communication networks [17,24–88].

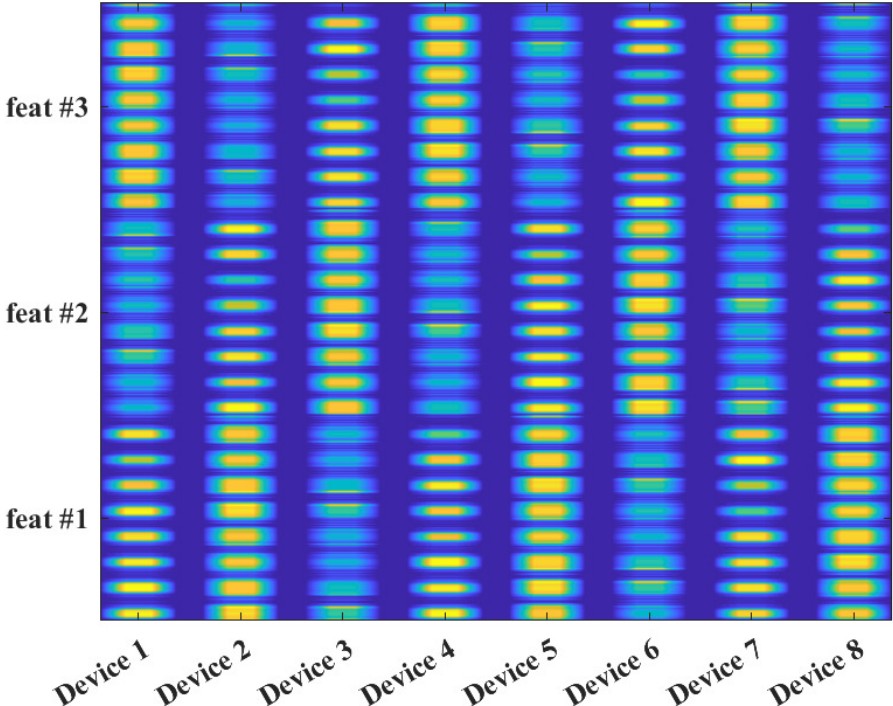

**Figure 1.** Representative illustration showing an average RF fingerprint for eight commercial emitters of the same manufacturer and model at a signal-to-noise ratio of 30 dB. This unique presentation has been designated RF-DNA because it highlights the "Distinct, Native, Attributes" or DNA found and exploited within an emitter's transmitted signals [33].

Despite the amount of research conducted within the SEI area, it remains largely a focus of academic efforts with little to no use as a security mechanism within operational wireless communication deployments. This is because such deployments must contend with (i) various operating environments and conditions whose dynamic nature obscures SEI exploited signal features, (ii) emitters that may enter and leave the network and change location or service Base Station (BS)/Access Point (AP), (iii) networks that consist of hundreds to thousands of authorized emitters let alone unknown emitters that are not part of the authorized set, and (iv) user devices that are constrained in compute, memory, or other resources essential to SEI steps or the process as a whole. Additionally, little focus has been placed on the essence of SEI regarding the origin or cause of a particular feature or set of features and how interactions within the emitter's RF front end impact them. Lastly, SEI research predominately treats the to-be-identified emitter as a passive source incapable of or unwilling to actively alter its signals or signal features to reduce SEI's effectiveness or defeat it altogether. These observations and the critical need for IoT security are the impetus for this comprehensive survey of the existing literature focused on chronicling recent advances that address one or more of the above observations and their pertinence to IoT security. We classify these existing works based upon the following criteria: (i) isolation of the source or sources of a feature or set of features within an emitter's RF front end, (ii) operating conditions, (iii) processes designed to impede or thwart SEI, (iv) SEI at scale, (v) size and availability of signal databases, and (vi) IoT imposed resource limitations or considerations. It is important to note that the focus of this survey is purely a technical one in terms of moving SEI from a proof-of-concept demonstration to a practical IoT security mechanism; thus, SEI's theoretical, managerial, or societal implications are not addressed as they are outside the scope of this survey. Instead, addressing these implications is left to future work(s).

The remainder of this paper is organized as follows. A comparison between prior surveys and this one is provided in Section 2, followed by Section 3 describing the process and criteria for selecting the reviewed literature. Section 4 describes what SEI is, the signal regions from which it can be learned or extracted, an equation expressing SEI feature variation within a signal, and SEI processes that leverage features associated with specific RF front-end components. Section 5 summarizes works that address the performance of SEI under alternate operating channel (a.k.a., non-Gaussian noise) and temperature conditions. Section 6 surveys papers that specifically look into adversaries focused on defeating or inhibiting SEI. Section 7 looks into performing SEI in large IoT deployments and across signal collections. Section 8 surveys publicly available signal data sets that can be used for SEI. Section 9 covers papers that develop SEI processes under the consideration of IoT imposed constraints (e.g., limited memory) and receiver-agnostic SEI to allow every receiver within an IoT deployment to be used by an SEI process. Section 10 covers challenges facing IoT-focused SEI that are not covered in the previous sections because there are too few papers to warrant individual sections. Section 11 concludes the survey.

## 2. Related Works

A comparison between the topics addressed in this paper and those of prior, related survey papers is provided in Table 1. The authors of [54] survey physical fingerprinting techniques for identifying mobile phones, including SEI works and other mobile phone components including but not limited to the camera, microphone, and display. Their survey also considers Medium Access Control (MAC) layer approaches and Physical (PHY) layer approaches, including RF fingerprinting. One essential contribution of the survey in [54] is adopting four fingerprint requirements borrowed from the biometrics domain [89]. These requirements are:

1. **Universality:** every emitter possesses the characteristics or features used to identify it.
2. **Uniqueness:** no two emitters have the same RF fingerprint or SEI exploited features.
3. **Permanence:** the RF fingerprint features are invariant to time or environmental conditions.
4. **Collectability:** the exploited features can be quantitatively measured.

**Table 1.** Comparison of the content of this survey versus previous surveys within the RF fingerprinting area.

| | Survey | | | | | | | | |
|---|---|---|---|---|---|---|---|---|---|
| | **This Paper** | **[54]** | **[90]** | **[91]** | **[92]** | **[93]** | **[94]** | **[95]** | **[96]** |
| Year | 2023 | 2017 | 2019 | 2020 | 2020 | 2021 | 2022 | 2022 | 2022 |
| IoT Motivated | ✓ | | ✓ | ✓ | | ✓ | ✓ | ✓ | ✓ |
| Handcrafted SEI | ✓ | ✓ | | ✓ | ✓ | ✓ | ✓ | ✓ | ✓ |
| DL-based SEI | ✓ | ✓ | | ✓ | | ✓ | ✓ | ✓ | ✓ |
| Operating conditions | ✓ | ✓ | | | | | | | |
| Threats to SEI | ✓ | | | | ✓ | | | | |
| SEI at scale | ✓ | | | | | | | | |
| IoT limitations | ✓ | | | ✓ | | | | | |

These requirements are included because they remain pertinent and, in some cases, unaddressed; thus, they inform our observations and analyses presented herein. While the authors of [91] survey Physical Layer Authentication (PLA), which encompasses SEI as well as channel-based authentication schemes that take advantage of the Jakes uniform scattering model to identify two communicating devices based on the unique channel response that exists between them [97]. Thus, the survey in [91] provides limited coverage of SEI. In particular, the authors note Deep Learning (DL) is an emerging PLA technique, the threat of SEI impersonation based on the work in [98], and IoT as the "next wave of technological evolution", which differs from our IoT-centric SEI survey. The authors of [92] focus on surveying SEI works based on the signal portions from which RF fingerprints are extracted, with the majority covering transient-based SEI. The authors of the survey in [93] also focus on PLA; thus, SEI only accounts for a small portion of the works surveyed. The SEI works covered in [93] are analyzed based on the signal region from which SEI features are extracted, the features used for SEI, and the classification processes. The survey presented in [96] covers the various machine-learning approaches used to detect and identify IoT devices. However, since the scope of the study is on machine-learning techniques, SEI and other IoT device detection and identification works are included. The survey presented by the authors of [90] focuses on Physical Layer Security (PLS) in fifth-generation (5G) wireless IoT networks. It primarily focuses on defining security threats and their purpose, categorizing them, and surveying 5G-specific countermeasures. The authors of [90] mention SEI as a PLS approach for authenticating legitimate IoT devices but do not survey SEI itself, which we do.

Our survey differs from those presented by the authors of [93–96] that use IoT as motivation or context for their surveys but do not use IoT as a lens through which to analyze the surveyed SEI papers, as we do herein. This survey aims to answer: "What technical gaps must be addressed for SEI to be a viable PLS solution for IoT deployments?" We answer this question by identifying these technical gaps by surveying papers whose methods and results address at least one of the following topics and sub-topics.

- Performing SEI under changing operating conditions such as alternate channels, Section 5.1, and environmental temperatures, Section 5.2.
- Investigating threats focused on reducing SEI's effectiveness or defeating SEI altogether, Section 6.
- Performing SEI as the number of emitters increases, Section 7.1, and using multiple collections conducted by the same receiver, Section 7.2.
- Identifying publicly available signal sets to standardize SEI process benchmarking, Section 8.

- Integrating SEI on resource-constrained IoT devices, Section 9.1, and using multiple receivers to collect signals from the same emitter or set of emitters, Section 9.2.
- Including additional literature that is relevant to IoT deployable SEI but does not fall into the research mentioned above, Section 10.

In addition to the overarching question stated above, each section ends with section-specific questions, observations, or comments that provide a brief conclusion to each section's referenced literature. The structure of each section and their technical topics and sub-topics are illustrated in Figure 2.

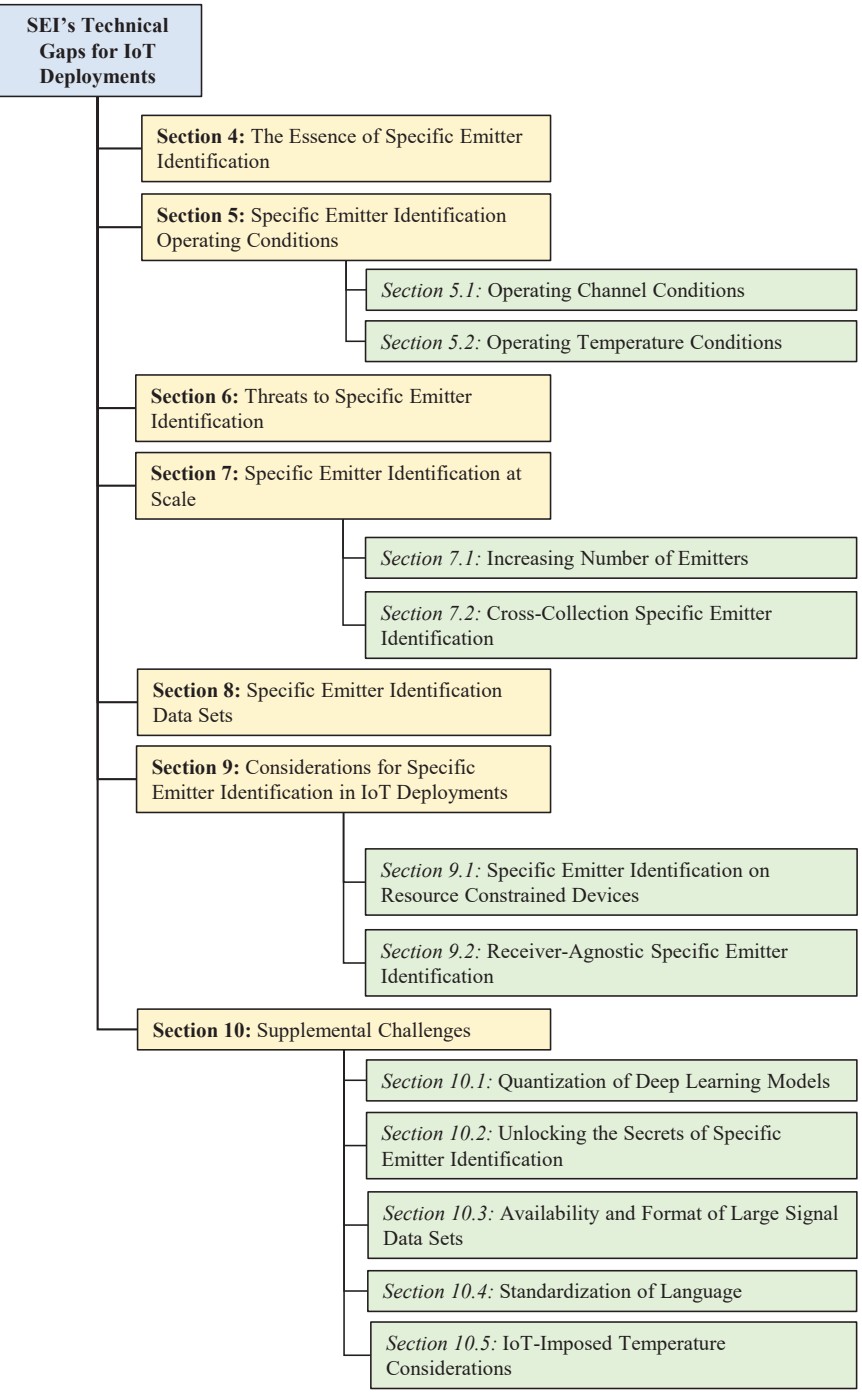

**Figure 2.** Illustrated structure of our survey listing the topics and sub-topics identifying SEI's technical gaps that must be addressed to make it a viable IoT deployable PLS solution.

## 3. Process for Identification of the Reviewed Literature

The publications reviewed in this survey were selected (i) by using search engines such as Google®, Google® Scholar®, or IEEE® Xplore®, (ii) from publications referenced in our prior works [37–41,48,50–52,78,80,99–107], and (iii) based on our continuous reading of current SEI publications and reviewing their cited references. When using a search engine, keywords were chosen based on the specific SEI topic or sub-topic being reviewed. Thus, keywords change from topic to topic and sub-topic to sub-topic, reflected in each section of this survey. For example, for Section 6, keywords included "attack", "threat", and "Eve". However, all searches included the keywords "Specific Emitter Identification", "SEI", "Radio Frequency Fingerprint", and "RFF".

## 4. The Essence of Specific Emitter Identification

SEI processes can be assigned to two general categories, which are designated here as (i) constellation-based and (ii) signal-based. Constellation-based SEI processes extract discriminatory emitter features (e.g., phase and amplitude shifts from the ideal point) from an individual constellation point, collection of constellation points, or distributions associated with the employed digital modulation scheme's two-dimensional, complex plane scatter diagram [98,108–110]. A digital modulation scheme's constellation is a byproduct of the demodulation process, and its corresponding points are generated by sampling each signal symbol at a specific time. The key to constellation-based SEI is to complete part or all of the demodulation process.

In signal-based SEI, discriminatory emitter features are extracted from the signal's discrete-time samples or their representation (e.g., spectrum). Signal-based SEI can be further subdivided into transient and steady-state-based approaches. Figure 3 illustrates the transient and steady-state portions of an IEEE 802.11a Wireless-Fidelity (Wi-Fi) signal. In transient-based methods, SEI is performed by extracting discriminating features from the temporary transitions that exist at the beginning and end of a transmission [31,111–113]. These turn-on and turn-off transients are very short in duration; thus, requiring high sampling rates in the gigabytes range [114] and making them difficult to detect and exploit for SEI as the channel conditions degrade (i.e., lower signal-to-noise ratios) [92,115].

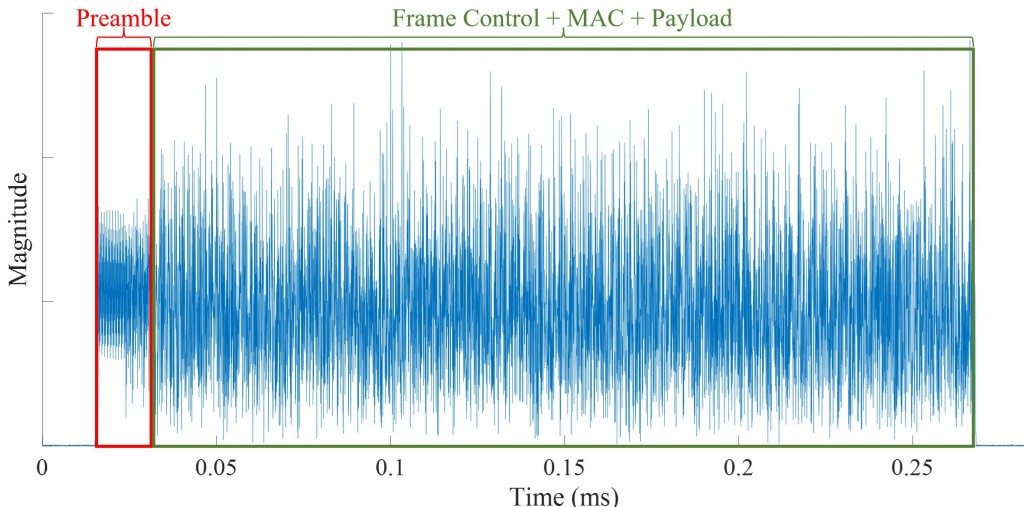

(**a**) Full signal with highlighted regions where SEI can be extracted or learned.

**Figure 3.** *Cont.*

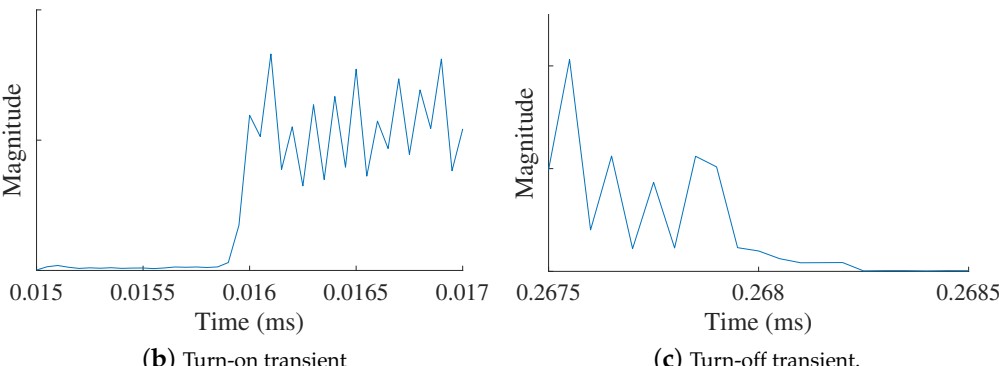

(**b**) Turn-on transient                                          (**c**) Turn-off transient.

**Figure 3.** Representative illustration showing the (**a**) full signal, (**b**) turn-on transient, and (**c**) turn-off transient portions of a transmitted signal from which signal-based SEI features can be extracted or learned.

Due to the limitations associated with transient-based approaches, other SEI works have focused on extracting the discriminatory features from the steady-state portion of the signal. The signal's steady-state portion is longer in duration and of higher energy than that of the transient; thus, making its detection much more effortless. Signal-based SEI processes can exploit a known, fixed sequence of signal symbols (e.g., the IEEE 802.11a Wi-Fi preamble) [41], the information-carrying symbols (i.e., the signal's payload) [57], or a combination of the two.

Regardless of the signal portion from which SEI features are extracted, the exploited features are often expressed as variations from the ideal signal's values. Specifically, variations in an ideal signal's amplitude, phase, and frequency, which are mathematically given as,

$$r(t) = A(t)[1 + \Delta A(t)] \exp\{j[2\pi(f_0 + \Delta f)t + \phi_0 + \phi(t) + \Delta\phi(t)]\} + n(t),\ 0 \leq t \leq T,\ (1)$$

where $A(t)$ is intentional amplitude modulation, $\Delta A(t)$ is unintentional amplitude modulation, $\phi_0$ is the initial phase, $f_0$ is the carrier frequency, $\Delta f$ is the Carrier Frequency Offset (CFO), $\phi(t)$ is intentional phase modulation, $\Delta\phi(t)$ is unintentional phase modulation, $n(t)$ is channel noise, and $T$ is the total duration of the signal [116,117]. It is the unintentional features that SEI exploits.

Researchers have investigated specific RF front-end components' contributions and impacts on SEI performance based on Equation (1). These investigations considered the role of the RF front end's Power Amplifier (PA), Analog-to-Digital Converter (ADC), Local Oscillator (LO) [48,88,118,119], and baseband Low-Pass Filter (LPF) [120,121].

The authors of [48] assess the impact of CFO on the SEI process using four Cisco AIR-CB21G-A-K9 Commercial-Off-The-Shelf (COTS) emitters. The experiments show how SEI performance is impacted when CFO is and is not present and when each emitter's CFO distribution is unique. The authors use Gabor Transform (GT)-derived RF-DNA fingerprints and a Multiple Discriminant Analysis/Maximum Likelihood (MDA/ML) classifier. The results show that identifying individual emitters is higher when CFO is present versus when it is not. The authors also show that SEI performance is highest when each emitter's CFO distribution is unique (a.k.a. non-overlapping with the distributions of all remaining known emitters). When each emitter's CFO distribution is modified to match Emitter #4's distribution, SEI performance is similar to when the CFO of all emitters is removed. When the CFO for each emitter is randomly drawn from the same distribution, SEI performance is consistent with the CFO removed case. The results in [48] show that SEI performance improves when CFO is present and uniquely distributed across each emitter.

The authors of [88] improve SEI performance by introducing the PAssive RAdiometric Device Identification System (PARADIS) technique. PARADIS uses an emitter's (i) frequency error (a.k.a., CFO), (ii) SYNC correlation, (iii) In-phase and Quadrature (IQ) offset

(a.k.a., DC offset), (iv) magnitude error, and (v) phase error. The SYNC correlation is calculated by correlating the received signal with an ideally generated one. The magnitude error is calculated by calculating the absolute distance between the signal's payload constellation point and the ideal constellation point location. The phase error is calculated as the difference in phase between the received and ideal constellations. To assess performance, the authors use a Support Vector Machine (SVM) or a *k*-Nearest Neighbors (*k*NN) classifier. The authors use the following performance metrics to compare the two networks: (i) average error rate, (ii) False Accept Rate (FAR), (iii) False Reject Rate (FRR), and (iv) worst-case similarity. The average error rate is the misclassification rate calculated across all emitters in the set. An emitter's FAR is the average rate at which the classifier incorrectly assigns a different emitter's signal to that emitter (e.g., Emitter A's FAR is the rate at which Emitter B is called Emitter A). An emitter's FRR is the average rate at which the classifier wrongly classifies the emitter as another emitter (e.g., Emitter A's FRR is the rate at which Emitter A is called Emitter B). The worst-case similarity is the highest rate at which an emitter is falsely assigned another emitter's features (e.g., if Emitter B is called Emitter A more than Emitter C is called Emitter A, the worst-case similarity reports the rate at which Emitter B is called Emitter A). The experiment uses one hundred thirty-eight ORBIT nodes that communicate using Atheros Network Interface Controllers (NIC) transmitting at 2.412 GHz and communicating via IEEE 802.11b Wi-Fi. Using the SVM classifier, the highest FRR is 10%, and the worst-case similarity rate is 16%. These rates increase when using the *k*NN, where the highest FRR increases to 62%, and the worst-case similarity increases to 40%.

The authors of [118] propose comparing an estimated CFO value to the actual received CFO to either grant or deny network access to an emitter. For an emitter to be granted access, it must fall within a given tolerance around the signal's CFO. Two factors determine this tolerance: (i) Doppler shift and (ii) receiver-incident Signal-to-Noise Ratio (SNR). The authors role-play a threat model where Alice is the true, trusted emitter, Eve is the adversary attempting to gain access to the network, and Bob is the authenticator that grants network access. Alice and Eve are modeled using the same Universal Software Radio Peripheral (USRP) Software-Defined Radio (SDR). The authors' experiment uses the preambles extracted from IEEE 802.11a Wi-Fi frames transmitted at a carrier frequency of 2.5 Ghz. At SNR values above 10 dB, Bob rejects Eve at a rate greater than 80%.

The authors of [120] introduce a method of validating the identities of trusted emitters using IQ Imbalance (IQI). IQI is the difference between the ideal, modulated, and real received constellation points. This is similar to the method introduced by the authors of [118]. The authors of [120] track IQI per emitter and assign a Gaussian distribution to account for perturbations caused by the wireless channel. The received signal's IQI is calculated and mapped to the known IQI. The received signal is assigned to the emitter whose distribution resulted in the highest likelihood value using the mapped IQI value. In the simulation, Bob can correctly authenticate Alice's identity at a rate of over 98% at an SNR of 18 dB and higher.

## 5. Specific Emitter Identification Operating Conditions

IoT devices operate in various environments; thus, effective SEI processes must contend with environmental and environment-induced effects that impact the identified channel or emitter. This section reviews SEI publications on operating channel conditions and emitter operating temperatures.

### 5.1. Operating Channel Conditions

Despite the amount of research within the SEI space, most of it has been conducted using simulated channel conditions. Those preponderances assume an Additive White Gaussian Noise (AWGN) channel model. Thus, this section reviews SEI works that employ alternate channel models to move SEI from a proof-of-concept demonstration to a realistic IoT security approach.

In [57,62,122–126], SEI performance is assessed using a multipath channel, but the specific characteristics of the multipath channel are unknown, unstated, or not disclosed.

The authors of [127] propose a semi-supervised SEI process built upon a Triple-GAN network. First, a representation network extracts features—from the received RF signals—that are used to train the Triple-GAN network, and feedback learning is used to assist the representation network in learning more discriminative features. Similarly, the authors of [128] present an unsupervised-learning SEI process that uses an Information maximized Generative Adversarial Network (InfoGAN) to train a discriminative model for emitter identification. The authors of [128] also apply RF Fingerprint Embedding (RFFE) by computing the bispectrum histogram from the received signal and integrating the resulting features into the InfoGAN's training process. The efforts in [127,128] evaluate SEI performance within a multipath environment modeled using a Nakagami-$m$ distribution. Although the value of the distribution shape parameter $m$ is chosen to simulate Rayleigh and Rician fading channels, the multipath fading channel is not described in detail. For instance, the authors in [127,128] do not specify critical channel configuration details such as the number of reflectors or paths and delay spread values. This information is essential when assessing and analyzing SEI performance for different channel lengths representing various multipath environments.

In [129], the authors present a DL-based device fingerprinting approach that leverages Multiple-Input Multiple-Output (MIMO) system capabilities and Space-Time Block Codes (STBCs) to mitigate the adverse effects AWGN and Rayleigh fading channels have on SEI performance. Since SEI features are distorted by Rayleigh fading channels, the approach in [129] exploits the MIMO system's multiple received signal streams to reconstruct a less-distorted version of transmitted signals, which are later used for model training and classification. Without knowledge of the channel state information, the transmitted signal is estimated at the receiver using two blind-source-separation and blind-channel-estimation algorithms, neither relying on pilot symbol-based estimation. The first algorithm fully calculates the channel matrix using Orthogonal STBC (OSTBC) properties and the received signal covariance matrix. The second algorithm attempts to partially estimate the channel by calculating a solution to a subset of the channel matrix up to some ambiguity. The receiver's reconstructed signals are expected to exhibit channel-immune characteristics when using DL models trained for channel conditions that differ from those present in the received signals before reconstruction. The presented approach's SEI performance is evaluated by adjusting the emitters' phase noise, CFO, and IQ gain imbalance hardware impairments within a fixed range. This allows the authors of [129] to simulate up to ten emitters with three antennas and Wi-Fi communication. The approach performs emitter identification via a Convolutional Neural Network (CNN) and signals that undergo varying Average Path Gain (APG) and Doppler shift changes. The authors leverage MIMO technology to mitigate channel effect variation on SEI performance. MIMO improves identification accuracy by 30% and 50% for AWGN and Rayleigh channels, respectively.

The authors of [130] propose an SEI process that exploits the different spectrum of adjacent signal symbols, known as the Difference of the Logarithm of the Spectrum (DoLoS), to extract RF Fingerprint (RFF) features that are robust to time-varying channels. The approach in [130] is motivated by the fact that during coherence time, the channel is considered stationary; therefore, two different symbols in one packet exhibit different SEI features but have the same channel response. DoLoS is calculated for two symbols of the same received signal to extract SEI features independent of time-varying channel effects. The proposed approach is evaluated using IEEE 802.11n signals collected from seven emitters that are located at twelve different locations to simulate different channel conditions. DoLoS calculates the spectrum difference between the IEEE 802.11 Wi-Fi preamble's Short Training Symbol (STS) and Long Training Symbol (LTS) sequences. The resulting DoLoS response trains and evaluates a CNN-based SEI process. The characteristic of the assumed time-varying multipath channel, including channel length and delay spread, are not described by the authors of [130]. The proposed SEI process is evaluated using seven

IEEE 802.11 Wi-Fi devices, twelve data collection positions, and two different environments. The presented SEI process achieves single- and cross-environment identification accuracies of 99.02% and 97.05%, respectively.

The authors of [131] propose an SEI process for wireless Orthogonal Frequency-Division Multiplexing (OFDM) device identification in time-varying channels. They attempt to cancel the time-varying multipath channel effects by extracting SEI features from the emitter's non-linearity and IQ imbalance using a Hammerstein system parameter separation technique. The algorithms in [131] are performed in three steps: (i) the Hammerstein system parameter separation technique estimates the emitter's non-linear model parameters as well as the multipath channel's Finite Impulse Response (FIR), (ii) the estimated FIR is used to obtain the best IQ imbalance parameter combination, and (iii) $k$NN is used to classify the estimated non-linear model and IQ imbalance parameters obtained in the first two steps. IQ imbalance and PA non-linearity values are set to simulate five emitters with minor differences. The proposed approach is evaluated within a Rayleigh fading channel with a maximum channel delay of nine samples. The authors of [131] do not specify the length of the Rayleigh channel or the delay spread corresponding to each path. The experimental results show that the extracted SEI features are stable, and the proposed authentication method is feasible.

The authors of [132] present an SEI process that exploits the IoT devices' PA non-linearity to generate environment-robust RFFs to improve SEI performance when signals are collected in the presence of time-varying multipath channels. The approach calculates the PA non-linearity quotient by performing element-wise division of the frequency response of two consecutive signals transmitted at two different power levels. This process mitigates the effect of the multipath channel, and the resulting quotients are used to train a CNN-based classification model. The authors use transfer learning to integrate RFFs from wireless environments to enhance SEI performance in the presence of multipath fast fading and Doppler. First, a base model is trained using channel effect-free RFFs from signals collected in an anechoic chamber. Then, the model is retrained using samples collected from indoor or outdoor environments to emulate moderate or severe multipath fading effects, respectively. A detailed description of the multipath channel, including the model type, is not provided by the authors of [132]. Compared to conventional DL and spectrogram-based models, the proposed PA non-linearity quotient and transfer learning-based SEI process improves indoor and outdoor environment performance by 33.3% and 34.5%, respectively.

The authors of [50,99] propose a channel estimator built using the Nelder–Mead (N–M) simplex algorithm to restore the SEI features before emitter identification when IoT signals are propagating in a multipath fading environment characterized by a Rayleigh fading model. The N–M estimator estimates the Rayleigh channel impulse response, which is later used by a Minimum Mean Squared Error (MMSE) equalizer to mitigate the multipath effects on the received signals. In [99], RFFs are generated from the equalized signals by computing the GT coefficients, subdividing the normalized magnitude of the GT coefficients into patches, and computing the statistics of variance, skewness, and kurtosis. The generated fingerprints are then used to train an MDA/ML classifier. The work in [99] analyzes the resulting SEI process using Rayleigh fading channels consisting of two, three, and five paths where the coefficient associated with each path changes for each IEEE 802.11a Wi-Fi frame. Further, Ref. [99] analyzes the effect of the Gabor patch size on the SEI classification performance. The authors of [52] present a DL-based SEI process that uses N–M-based channel estimation and MMSE equalization and CNN pretraining to improve SEI performance in IEEE 802.11a Wi-Fi indoor Rayleigh fading channels. The work in [52] performs SEI using Rayleigh channels comprising up to seven paths. Further, the authors of [52] investigate different IEEE 802.11a preamble representations, including a one-dimensional IQ representation and a two-dimensional time–frequency image generated from the GT coefficients. To reduce the CNN's size, augment the training data set, and improve the CNN's capability to extract shift-invariant SEI features, the authors of [52] adopt data partitioning that slices the preambles' one-dimensional and

two-dimensional representations into shorter sequences and sub-images, respectively. Finally, the authors of [52] use a Convolutional AutoEncoder (CAE) to pre-train the CNN to improve model convergence and SEI performance. When compared to traditional feature-engineered SEI processes, the experimental results in [52] demonstrate that using N–M channel estimation and equalization with a CAE-initialized CNN and data partitioning improves SEI identification performance by 9% at an SNR of 9 dB.

The authors of [100] present two semi-supervised learning-based approaches to restore (a.k.a., correct) SEI features when IoT signals are corrupted by Rayleigh fading channels. The first approach aims at estimating a generative function that compensates for multipath channel effects by training a Conditional GAN (CGAN) network using (i) signals that have been corrupted by multipath channels, (ii) the signals originally transmitted without multipath, as well as (iii) the label associated with each signal. Combining the label with the signal's representation enables the CGAN's generative model to compensate for the multipath effects while preserving emitter-specific SEI features. A separate CNN model is trained to classify the CGAN corrected (a.k.a., equalized) signals. The received multipath signal is combined with all possible labels at the generator's input, and the signal is assigned to the label that achieves the highest confidence score at the output of the CNN classifier. The second approach uses a Joint CAE and CNN (JCAECNN) architecture comprising multiple decoder heads and a single classification head to compensate for multipath effects while preserving SEI discriminating features. The JCAECNN's multiple decoder and classification heads are jointly trained to decompose the multipath corrupted received signal into its original delayed and weighted versions. Each of the delayed and weighted versions is classified separately using a CNN model, and a final decision is made by combining all decisions using a highest-vote scheme. Inspired by the fact that the Rayleigh channel's path coefficients decay exponentially, the authors of [100] introduced exponentially decaying loss weights that improve the overall SEI performance. The CGAN and JCAECNN-based SEI processes improve SEI performance by 10% when extracting emitter-specific features from signals collected under a Rayleigh fading channel comprising five paths or reflections, which is superior to prior SEI processes.

The authors of [133] present Channel Robust Representation Networks (ChaRRNets) for SEI when signals are received under multipath conditions. In particular, the authors build a CNN whose convolutional layers are equivariant and invariant to the channel's frequency response to mitigate or eliminate adverse multipath channel effects on the SEI process. The authors create these equivariant and invariant CNN layers using the Albein Lie group representation of the complex-valued signals (a.k.a., the IQ samples). They aim to develop a CNN that can handle cases in which it is trained using signals collected under Rayleigh fading and tested using signals collected under Rician fading conditions. The authors assess ChaRRNets using signals that have simulated SEI features applied to them using an Infinite Impulse Response (IIR) filter and signals transmitted by real-world emitters. It is unclear the specific features that are imparted by the IIR filter; however, the SEI features do seem to be unique for each emitter and are unchanged across each emitter's simulated signals. Although such an approach is suitable for validating ChaRRNets, it does not indicate real-world emitters whose SEI features change from one signal to another, as shown by the work presented in [48]. The real-world signals used to test ChaRRNets are collected on another day than those used for training to ensure the channel conditions differ. The manufacturer and model of the real-world emitters are not provided, which makes it unclear if this is SEI at the serial number, model, or manufacturer level. The first is the most challenging case of SEI, and the last is the easiest. The authors also perform "CFO augmentation" but do not explain what it is, how it is implemented, or a citation describing it. Unfortunately, CFO has been shown to bias DL-based SEI processes, making them susceptible to attack [48,101,117]. ChaRRNets outperforms a conventional complex-valued CNN with an average classification performance of 65.5% versus 25.6% when training ChaRRNets using both days of real-world, "in the wild" collected signals. The authors do not present classification performance for each emitter.

Technical Gaps—Channel Conditions

Non-AWGN channel model usage must continue; however, further research is needed. In particular, any deployed SEI model must be adaptable to changing channel conditions but in such a way that any change to the model does not reduce or eliminate the effectiveness of the learned features used to discern one emitter from another. Future SEI research must include modeling and simulation using clearly defined variables and channel models, testing in controlled environments, and, eventually, operational IoT networks to assess the efficacy of any SEI process honestly. Fortunately, some platforms can be used for such purposes. One example platform that could be used is the Platform for Open Wireless Data-driven Experimental Research (POWDER) [134].

*5.2. Operating Temperature Conditions*

IoT devices operate in various environments, many impacting device operating temperature. An example is IIoT devices that can be deployed in manufacturing settings associated with extreme temperatures due to the manufacturing processes themselves (e.g., steel production and processing) or the conditions under which a manufacturing process is running (e.g., non-temperature-controlled building). Therefore, SEI processes must continue to provide effective and accurate emitter identification regardless of the temperature(s) under which the emitter(s) operate. The SEI publications surveyed under this section focused on the environmental temperature impacts on SEI and not emitter-connected operating temperature(s). The former is not new to the SEI community [135] but has not received much attention [114,136,137]. The latter is addressed in Section 10.5 of this survey.

The Temperature-aware Radio Frequency Fingerprinting (TeRFF) approach is presented by the authors of [114]. TeRFF uses receiver-measured CFO to discriminate one emitter from another and is motivated by the fact that this LO-dependent feature is directly influenced by the temperature of the RF front end [138]. The authors of [114] address this temperature dependency by training a naïve Bayes classifier for each of the eight discrete temperatures within the range of 26 °C to 33 °C (i.e., there is 1 °C difference between consecutive temperature values). The authors adopt this approach, assuming that temperature-independent SEI features are difficult to learn. An emitter's CFO value is estimated whenever that emitter operates outside the designated temperature range. To eliminate the need to compare every signal or RFF with each of the eight naïve Bayes classifiers (i.e., one for each temperature), the authors of [114] require each emitter to transmit its current operating temperature via Internet Control Message Protocol (ICMP). This provides an opportunity for exploitation by nefarious actors because it provides information (a.k.a., operating temperature) about the IoT device. Additionally, the ICMP requirement adds complexity to the SEI process at the loss of its passive nature, assumes the capability is organic to the emitter or device, and consumes more onboard resources while increasing processing times. The last two are essential factors to consider if the TeRFF approach is deployed in IoT or IIoT deployments because they may require the integration of additional functionality or modification of the device that can limit the associated IoT devices' operational longevity. Lastly, TeRFF uses CFO as the emitter-specific feature, which is an easily manipulated feature that can be exploited by nefarious actors [48].

In [137], the authors collect signals from emitters operating at temperatures of −5 °C, 10 °C, 25 °C, and 40 °C. SEI is performed using the CNN architectures of ResNet50 and InceptionV3, which learn emitter-specific features from their signals' transient region detected using a unique technique the authors coined the "double sliding window method". The SEI performance is high when CNN training and testing are performed using transients extracted from signals collected at a single temperature. However, SEI performance is seriously degraded when the training set of transients differs from those comprising the testing set (a.k.a. cross-temperature SEI). To address this performance degradation, the authors construct "blended" training and testing data sets containing equal transients collected at each of the four temperatures. Despite this, SEI performance remains low when

using the "blended" data sets, with an average percent correct accuracy of 45% and 52% when the InceptionV3 and ResNet50 classifiers are used to identify emitters operating at −5 °C, respectively. This represents the lowest average classification accuracy using the "blended" data set.

The authors of [102] investigate environmental temperature impacts on preamble-based SEI for temperatures ranging from −40 °C to 80 °C. The authors collect preambles transmitted by four HackRF One SDR emitters in an environmental chamber. The emitters transmit an ideally generated IEEE 802.11a Wi-Fi frame at a carrier frequency of 2.45 GHz over a wire to a receiving USRP B210 that samples the received signals at a frequency of 40 MHz. The B210 is placed outside of the environmental chamber. Collections are performed at the previously stated temperature range in steps of 10 °C for thirteen different environmental temperatures. A total of 1,000 preambles are collected from each emitter at the designated temperatures. The authors employ four SEI processes: (i) MDA/ML, (ii) Principal Component Analysis (PCA)/$k$-Means, (iii) CNN, and (iv) Long Short-Term Memory (LSTM). The MDA/ML and PCA/$k$-Means approaches use GT-derived RFFs, the CNN uses the raw image of the GT normalized magnitude, and the LSTM uses the magnitude and phase of the preamble's IQ samples. When training and testing each network at ambient (a.k.a., 20 °C), the MDA/ML network has an average classification accuracy across all devices of greater than 98%, the PCA/$k$-Means has an accuracy of 25%, and the CNN and LSTM have an average accuracy of over 90%. When classifying temperatures below and above ambient, the average classification performance decreased. The authors then trained each network on a thirteen-chose-two combination of environmental temperatures for seventy-eight combinations. The results show that the highest average classification performance across all temperatures is achieved when the networks are trained on the extremes of the temperature ranges (e.g., the highest-performing LSTM network was trained at −30 °C and 70 °C). The results also show that blind SEI performance is higher at sub-ambient temperatures than at temperatures above ambient. The authors conclude that (i) the SEI processes can generalize the emitter's RFFs better when given the extremes of the temperature range and (ii) the emitter's RF front end performs more consistently at sub-ambient temperatures.

Technical Gaps—Operating Temperature

In addition to the challenge described in Section 10.5, the papers surveyed in this section show that an emitter's operating temperature remains an open problem in need of further investigation and solution development to ensure viable SEI-based security within IoT deployments.

## 6. Threats to Specific Emitter Identification

As with any security approach, one must remain aware of the fact that adversaries will endeavor to find ways to circumvent it. This is no different for SEI-based security approaches; thus, this section summarizes published SEI works that investigate weaknesses or specific techniques that adversaries can exploit to defeat or diminish the effectiveness of an SEI process.

The work in [98] is one of the earliest—if not the first—work to investigate SEI threats. The authors examine threats that employ feature and signal replay to defeat modulation and transient-based SEI techniques. It is worth clarifying that the modulation feature replay targets constellation-based SEI, which leverages modulator imperfections such as IQ origin offset, wireless frame magnitude and synchronization (SYNC) correlation errors, or wireless frame phase errors. Interestingly, the effectiveness of the threat's attack is nearly 100% when modifying and replaying modulation-based features. The authors also note that when a "high-end Arbitrary Waveform Generator" (AWG)—capable of sampling at 20 GHz—replays the signals of the targeted emitter, the transient-based SEI process struggles to distinguish the adversary from the emitter that the adversary targets. However, these results are generated when the AWG and SEI process receiver is connected via wire;

thus, the authors note that the success of this attack will be diminished due to channel and antenna impacts. The authors do not present results validating this statement. They also conduct all their experiments at a high SNR that is not specified, but the distance between antennas is. It is also unclear whether the SEI process and the adversary use the same receiver, but they seem to be the same. Using the same receiver may give the adversary an unrealistic advantage because receivers impart signal coloration (see Section 9.2). However, further research is needed to determine whether this favors the adversary. Lastly, all emitters—adversary, SEI process, and targeted/legitimate—are the same model USRP SDRs. This also favors the adversary because its RF front-end imperfections are most similar to those of the targeted emitter. Such a case is unrealistic in IoT deployments because IoT devices are not likely to be implemented using SDR due to cost constraints. Thus, future research must consider cases where the adversary uses an SDR but the targeted emitter is a COTS device.

The works in [48,101,117] investigate circumventing an SEI process through an adversary's ability to exploit a specific signal—and sometimes SEI—feature. The works in [48,117] look into an adversary's exploitation of CFO. CFO is the difference between the frequency of the transmitter's LO and the receiver's LO. In communication systems, CFO is estimated and corrected by the receiver before demodulation. CFO is an SEI-exploitable discriminating feature for emitters whose corresponding CFO distributions are unique amongst a set of known emitters [48]. An illustration of this is presented in Figure 4a. The CFO distributions of Emitter #1 and Emitter #8 overlap one another slightly, suggesting a higher level of emitter separation by the SEI process. In contrast, those of Emitter #3 and Emitter #4 overlap one another completely. This second case suggests that the SEI process will struggle to separate these two emitters when CFO is present in their signals. The results presented in [48,117] show that an adversary can easily monitor the CFO behavior of another emitter and then manipulate its LO to obtain a similar CFO behavior—within its transmitted signals—that diminishes or even thwarts the SEI process' ability to distinguish the adversary emitter from the emitter targeted by the adversary.

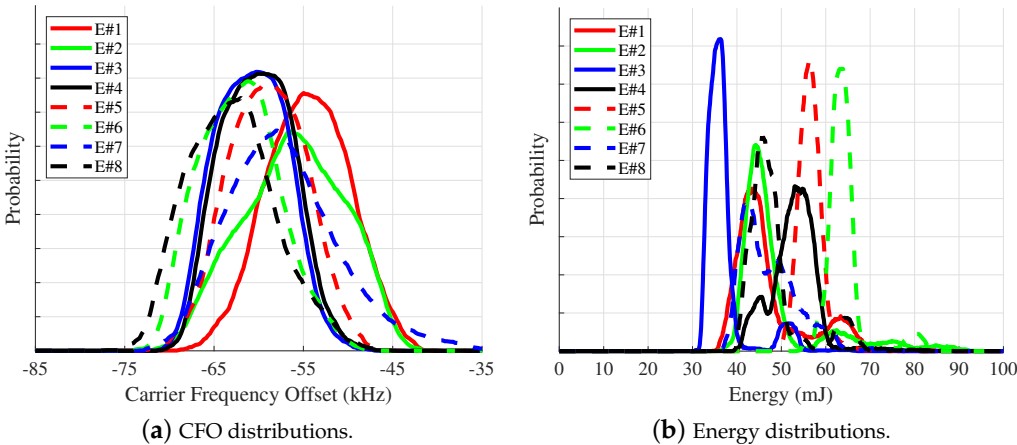

**Figure 4.** Probability Mass Functions (PMF) for the receiver-incident Carrier Frequency Offset (CFO) and energy measured from eight Commercial-Off-The-Shelf (COTS) TP-Link Archer T3U USB Wi-Fi emitters, denoted using E#.

The authors in [139,140] investigate an adversary's ability to spoof the SEI features of another emitter with the intent of posing as that emitter to (i) emulate a primary user within a cognitive radio network, (ii) circumvent a signal authentication system, and (iii) gain unauthorized access into a protected network. The authors investigate replay and Generative Adversarial Network (GAN)-based signal spoofing. Assessment includes SEI performance when the adversary transmits without actively spoofing the SEI features of another emitter but transmitting random signals. The SEI features targeted for spoofing are power and phase shift; however, the authors do not specify whether these features

manifest in the signal or constellation domain. Against DL-driven SEI, the results presented show that the adversary achieves a spoofing attack success rate of 7.89% when transmitting random signals (a.k.a., naïve spoofing), 36.2% for the replay-based spoofing attack, and 76.2% for the GAN-based spoofing attack [139]. In [140], the authors extend their work of [139] to include MIMO configurations of two and four antennas employed by the adversary or the receiver conducting SEI. The GAN-based spoofing attack can achieve an attack success rate of 88.6% and 100% when the adversary employs two and four antennas, respectively. The authors note that the adversary's attack success rate increases with more transmitter antennas but decreases when the SEI process' receiver employs more antennas. Thus, the success of the spoofing attack seems to improve as the adversary employs more sophisticated methods (e.g., GAN and MIMO.). Despite the value of the work presented in [139,140], some key observations must be considered for the threat it poses to SEI-based IoT security approaches. First, all of the research appears to be conducted using simulation because the authors never provide any hardware specifics (e.g., manufacturer and model of the emitters.) Thus, the efficacy of the presented approach in spoofing the SEI features of an actual emitter (i.e., hardware implementation) will need to be explored. Second, the authors use power as one of the exploited SEI features, which faces the same issues as those faced by SEI systems relying upon CFO or energy [48,101,117]. Third, the adversary's receiver must be placed near the SEI process receiver. Proximity was necessary to ensure the two receivers experienced the same or similar channel conditions. This appears to be an unrealistic requirement within operational IoT deployments. In [140], the authors assess the GAN-based spoofing attack's success rate when its location differs from that of the SEI process' receiver. As the adversary's receiver moves farther from the location of the SEI process' receiver, the less effective the GAN-based spoofing attack becomes, dropping from 100% to 56.2%. Fourth, the SEI spoofing results presented in [139,140] require the adversary to employ a transmitter and receiver that are not co-located; thus, requiring additional coordination, management, and resources that may increase the adversary's chances of being detected. Lastly, the authors do not specify the number of emitters that can be exploited by the adversary and omit spoofing performance against individual emitters. This is important when considering that an adversary will exploit the "weakest link" in any security system; thus, the adversary will look for the emitter whose SEI features most resemble its own organic SEI features or are spoofed with the greatest effectiveness.

The intention of the work presented in [108] is to collaboratively manipulate—using DL—the constellation-based SEI features (e.g., offset between emitter mean and ideal constellation point) of an emitter (a.k.a., Alice) to make it easier for the receiver (i.e., the device performing SEI) to identify Alice from a collection of emitters that are of the same manufacturer and model as Alice. The goal is to overcome SEI limitations within large populations of emitters. The authors assume the existence of a "feedback channel" between Alice and the receiver (a.k.a., Bob) during the SEI training phase. The authors also assess their DL-driven SEI feature manipulation approach's performance using an adversary emitter—designated as Trudy—capable of spoofing Alice's SEI features with increasing sophistication. A conclusion that can be drawn from the work in [108] is that the most sophisticated version of Trudy is capable of adding confusion into the SEI decision that subsequently forces Alice to modify its SEI features at the expense of degrading communications performance (i.e., poorer Bit-Error-Rate, BER) between Alice and Bob. These results highlight a drawback to constellation-based SEI, which is the fact that the constellation is two-dimensional. The two-dimensional nature limits the amount of variability—intentional and unintentional—that can exist between emitters while simultaneously maintaining communications performance. It is this constrained variability that may be exploited by an adversary or group of adversaries, especially as the modulation scheme changes (e.g., going from 8-Quadrature Amplitude Modulation (QAM) to 32-QAM), because it reduces the distance between constellation points.

In [141], the authors present a Deep Neural Network (DNN) architecture—designated as *FIRNet*—that is purpose-built to attack wireless DL networks by spoofing the signal

features of another emitter. Wireless channel impacts, the adversary's signal features, and a threat model are considered when assessing *FIRNet*'s efficacy. Using a threat model is key to assessing security countermeasures with specific objectives and vulnerabilities in mind and is often missing in SEI publications. The authors also consider both targeted and non-targeted adversarial attacks. In a targeted attack, the adversary attempts to make emitter $E_i$'s signals be identified as originating from emitter $E_j$. In contrast, the adversary attempts to make emitter $E_i$'s signals identified as originating from any emitter other than $E_i$ in a non-targeted attack. The authors' approach considers two adversarial scenarios. The first assumes the adversary has unrestricted access to the DNN performing SEI and is designated as a "white box" scenario. The second scenario is a "black box" scenario in which the adversary does not have unrestricted access to the SEI-performing network. However, the black box scenario does assume the adversary has access to the SEI network's final layer activations. Additionally, *FIRNet* is trained with the SEI process' trained DNN integrated into its training processes. Lastly, *FIRNet* is evaluated using five "nominally identical" USRP N210 SDRs. The fact that all emitters are of the same manufacturer and model gives the adversary the best chance of spoofing the other emitters' SEI features. This is because emitters of the same manufacturer and model are constructed using the same components, sub-systems, and systems; thus, there will be a greater similarity between the adversary's SEI features and those observed or learned from the signals of the remaining emitters. The adversary's use of an SDR is reasonable, considering the level of control it needs over its emitter to implement an attack. However, using SDRs as user equipment does not represent the typical, low-cost emitters in IoT devices. Therefore, *FIRNet* or similar adversarial approaches need to be assessed using actual IoT device emitters or ones that align with those used in IoT deployments to evaluate the threat's effectiveness to SEI-based security approaches fully.

The authors of [142] leverage an adversarial machine-learning process—designated as Radiometric signature Exploitation Countering using Adversarial machine-learning-based Protocol switching (RECAP)—focused on countering SEI. RECAP attempts to confuse an SEI process by (i) having multiple emitters transmit the same message simultaneously so their signals are synchronized in time and (ii) employing protocol switching at the link and physical layers. Link layer protocol switching is implemented by switching each device amongst its emitter's supported protocols. Physical layer protocol switching is achieved through distributed beamforming in which the transmitting emitters' signals are combined to form a new set of SEI features. Adversarial machine learning selects which link layer protocol is used to transmit the next message and which devices participate in distributed beamforming. RECAP is an elaborate SEI threat that requires a lot of coordination and adds complexity to any device employing it. However, RECAP represents a strong adversary that warrants further consideration and study within the SEI research community.

The authors of [143] introduce *RF-Veil*, an algorithm that randomizes an emitter's phase errors intending to make SEI robust against impersonation attacks while obfuscating the emitter's SEI features from non-cooperative receivers (e.g., eavesdroppers). Although the authors present *RF-Veil* as an SEI-based security enhancement, it is not difficult to imagine that nefarious actors may employ it to obfuscate their own SEI features to prevent, hinder, or circumvent legitimate SEI-based security processes. The use of *RF-Veil* by nefarious actors is not considered by the authors of [143], so future research should explore such an application. Based upon the work in [143], there are a few things to consider. First, the authors' work is focused on Channel State Information (CSI)-based fingerprinting. Although CSI-based fingerprinting is outside the scope of this survey, the work in [143] does raise the question as to how *RF-Veil* or a similar approach impacts the effectiveness of signal and constellation-based SEI. Second, the authors focus solely on obfuscating an emitter's identity by randomizing its phase information; thus, focusing on a singular feature. Does a phase-focused approach create a vulnerability that opens it up for exploitation or attack, as highlighted by other SEI works that considered a singular feature [48,101,117]? What about SEI processes that leverage multiple features? How is their effectiveness impacted?

Lastly, the authors of [143] do not consider degrading SNR or other channel impairments (e.g., multipath, interference, etc.). Such channel conditions must be considered when determining whether *RF-Veil* is used for good or ill.

The research presented in [144] uses adversarial learning to manipulate the signals' IQ samples in real time using online learning. The adversary uses only binary feedback—from the SEI process—to determine if its IQ manipulations are effective or not in spoofing the signal-based identity of one of the $N_e$ emitters. The weights and biases of the adversary's generative network are adapted based on the SEI process' response. The authors in [144] assess their approach using both simulations and a hardware testbed constructed using eight Analog Devices Active Learning Module (ADALM) Pluto SDRs. Although the results presented in [144] can "fool" the SEI process at a high rate (e.g., 90% or higher at signal-to-noise ratios of 15 dB and above), the authors conclude that the adversary does not learn the SEI features of the targeted emitter. Thus, the approach's effectiveness may have more to do with the classification. Classifiers perform a one-to-many comparison between the input sample (a.k.a., signal or its representation) and each class model representing an emitter within the known set. The input sample is assigned to the class whose model results in the "best" fit (e.g., smallest distance, largest probability, etc.). However, this class assignment is made even when the fit is poor. It is also important to note that all emitters are SDRs of the same manufacturer and model, including the adversary's SDR. Using SDRs of the same manufacturer and model represents a best-case scenario when it comes to SEI feature manipulation because there is less feature variability amongst emitters that only differ in the serial number. Therefore, it seems to provide the adversary with the greatest chance of fooling the SEI process because its own SEI features should be inherently similar to those of the emitter being spoofed and require the least manipulation. The authors do not (i) test the spoofing effectiveness of the adversary's organic SEI features (i.e., the adversary is not manipulating its own SEI features) or (ii) assess SEI performance when the adversary's emitter is not of the same manufacturer and model as the spoofed emitter. The adversary's use of an SDR is intuitive because an SDR grants the SEI manipulating algorithm direct access to its IQ channels/connections before the analog transceiver (a.k.a., the RF front end). However, it is unlikely that IoT devices will employ SDRs due to Size, Weight, and Power-Cost (SWaP-C) constraints. Therefore, future SEI work should consider the adversarial approach presented in [144] while considering some or all of the challenges highlighted here.

As with CFO, the work in [101] shows similar results and vulnerability regarding the energy distribution associated with an emitter's transmitted signals. In other words, when an emitter's signal energy distribution is unique, the SEI process can easily discriminate that emitter from all other emitters within the known set. However, an adversary can easily manipulate the energy used to transmit its signals to mimic the energy distribution of another emitter. Figure 4b provides an illustration of this in which the adversary—designated as Eve (E#1)—has a signal energy distribution that underlies that of another, known emitter (a.k.a., Alice, E#2); thus, allowing Eve to be identified as Alice by the SEI process [101]. SEI processes tend to learn the easiest feature or set of features that allow discrimination of one emitter from another within a set of known emitters. The work in [48,101,117] all demonstrate SEI processes that primarily exploit a single feature (i.e., CFO or energy) to facilitate emitter discrimination but at the cost of making them susceptible to adversaries that are capable of taking advantage of this singular vulnerability.

Similar to the work in [143], the authors of [145] present a CFO obfuscation technique to prevent adversaries from performing SEI of Bluetooth Low Energy (BLE) emitters. However, we discuss the work in [145] because adversaries can mask their CFO features or behaviors. The work in [145] differs from the work presented in [143] in that the transmitter intentionally obfuscates the CFO by adding a randomly selected value to it and applying it to the entire BLE transmission. Additionally, the authors of [145] increase CFO variation across signals by running the transmitter's Phase-Locked Loop (PLL) in a "temporarily unlocked" state that has the PLL's Voltage Controlled Oscillator (VCO) operating in an

open loop configuration. The VCO's open loop configuration allows the frequency to drift, adding an unpredictable amount of CFO to the signal above what was intentionally added. The result of this CFO obfuscation approach necessitates persistent observation and measurement for twenty-eight hours or more—of the obfuscating emitter's signals—before the corresponding CFO distribution's statistics can be learned. The authors provide a circuit design for a Wi-Fi and BLE emitter incorporating their proposed CFO obfuscation approach. The work in [145] further validates that CFO is a poor SEI feature due to the ease at which adversaries can passively learn CFO—even if it takes a day or more—as well as manipulate their own. However, a key value of the work presented by the authors of [145] is a circuit designed to defeat or hinder SEI. Future research needs to consider a similar approach or approaches focused on other SEI-exploited features as expressed by Equation (1).

The authors of [146] take a different approach to SEI attacks, built on the observation that DNNs are easily tricked by perturbed input data. Such perturbations have been shown to reduce DNN effectiveness or cause the DNN to fail altogether [147]. The authors present attack and defense scenarios. For the attack scenario, the adversary can select from one of four perturbations, Fast Gradient Sign Method (FGSM) [148], Projected Gradient Descent (PGD) [149], and Carlini & Wagner (C&W) adversarial attacks [150], but results are presented using only the FGSM perturbation approach. The authors suggest that the perturbations can be added to the IQ samples of the adversary's signals before the Digital-to-Analog Converter (DAC); however, the authors do not add the perturbations before the adversary's DAC and instead add them to the received signals (i.e., after the signals have been transmitted, traversed the channel, and been received). Thus, it is unclear how easily the adversary can implement the perturbation, the effects the channel and the RF front ends of the adversary and receiver would have on the perturbation, and whether or not such perturbations would negatively impact the demodulation process. The latter observation is important because it is safe to assume that an adversary's activities would not cease once it has been incorrectly granted access to the IoT system/infrastructure. Thus, if the perturbations cause sufficient signal distortion or bit errors to prevent demodulation, then the value of the attack is minimized or rendered useless. Nonetheless, the adversary's FGSM-based perturbation attack significantly reduces the SEI processes' ability to separate the adversary from the authorized emitters. It does so without the need to collect another emitter's signals or knowledge of the SEI exploited feature(s). However, the authors state that the adversary's perturbations leverage characteristics of the targeted DNN but do not provide specific details, what happens if the adversary does not have this knowledge, or how this knowledge is obtained. The authors counter the adversary's attack through their defense scenario, leveraging adversarial training to improve DNN-based SEI performance by 60% or more. It is important to note that the SEI process does know the adversary through collected unperturbed signals. The authors do not consider DNN performance—with and without adversarial training—when knowledge of the adversary's unperturbed signals is removed.

*Technical Gaps—Threats to SEI*

Traditionally, SEI research has assumed the exploited features are difficult to mimic [53,81–83,110] and the emitter being identified is passive or benign. However, the works reviewed in this section indicate that this is no longer the case and that ongoing SEI research must consider and contend with strong adversaries. A strong adversary is defined here as one that actively attempts to hide its own SEI features or manipulate them to subvert or impede SEI-based security approaches. Thus, it is imperative to assess future SEI processes with strong adversaries in mind while removing weaknesses that provide adversaries with points of exploitation (e.g., CFO, signal energy, feedback in the way of acknowledgments, etc.). In addition, SEI must be employed with other security mechanisms to develop a "defense-in-depth" (a.k.a., layered) IoT security approach built on the tenets of prevention, detection, and response, which are essential to any security strategy.

## 7. Specific Emitter Identification at Scale

This section summarizes publications focused on assessing SEI's efficacy in providing robust and effective PHY-based security in the face of a large number of emitters and the identification of those emitters across collections.

### 7.1. Increasing Number of Emitters

IoT deployments can and do consist of tens to hundreds of individual devices; thus, any SEI-based security approach must maintain its effectiveness while contending with large deployments. Most SEI investigations have focused on proof-of-concept demonstration; thus, SEI's effectiveness within a large set of emitters (e.g., fifty or more) has received little attention until recently. This change has been partly driven by DL's demonstrated success in natural language and image processing and facial recognition in the presence of large training data sets, such as the MNIST data set of handwritten digits, under ever-increasing amounts of data. This has been exacerbated by recent pushes to leverage DL for spectrum management [151] and emitter identification [152] by the Defense Advanced Research Projects Agency (DARPA); thus, the remainder of this section focuses on summarizing SEI works that consider thirty or more emitters.

The authors of [57] utilize a tuned FIR filter and CNN to identify five, ten, fifteen, and twenty USRP emitters. The FIR filter taps are optimized during training along with the CNN's weights. The number of filter taps is set to three, five, and ten. Accuracy is measured in two ways: (i) Per Slice Accuracy (PSA) and (ii) Per Batch Accuracy (PBA). PSA is the average accuracy when the network infers a single signal. A batch is a set of consecutive slices, giving the network a longer sequence of IQ samples to learn from. This helps the network by allowing for more temporal features to be visible. For all numbers of emitters in the set, the SEI performance increases roughly 20% when the optimized FIR is introduced. When the FIR filter is implemented, the ability of an adversary to mimic a trusted emitter is decreased from 10% to approximately 1% in a set of twenty USRP emitters. When the number of emitters is increased to one hundred, the PBA increases by 30% when using an FIR filter consisting of ten taps. When training and testing on a set of five hundred Automatic Dependent Surveillance-Broadcast (ADS-B) emitters, the highest performance improvement is 82% using PBA with a set of three batches, one hundred samples per slice, and a ten tap FIR filter. Similarly, the highest performance improvement is 55% when using the same parameters as the ADS-B set.

The authors of [153] use the Differential Constellation Trace Figure (DCTF) and Differential Interval (DI) to identify fifty-four Zigbee emitters. The DCTF is used for its ability to highlight differences unique to each emitter in the set. The authors propose using the DCTF's amplitude and phase. The two representations are then feature-reduced using the Gini importance or Relief-F algorithm. The reduced feature sets train a random forest classifier and a *k*NN. When classifying emitters at an SNR of 10 dB and the DI is set to 80, the random forest had a higher accuracy than the *k*NN classifier with both the Gini importance and Relief-F algorithm reduced DCTF features. When classifying at a range of DI values, the minimum value to maintain a classification accuracy above 95% at an SNR of 5 dB is 40. When comparing the classification accuracy of the different methods at 5 dB, the amplitude and phase representation accuracy with no feature reduction was 97%. This method had the same accuracy as when the phase-only representation was used without feature reduction and when the amplitude and phase representation were used with feature reduction. The performance decreases to 66% when only amplitude is used without feature reduction. Though the authors do not draw this conclusion, the results in [153] reiterate that the signals' phase representations tend to be more robust against noise than amplitude representations.

The authors of [103] investigate improving SEI performance by removing the intentional structure from the received signals transmitted by eight, sixteen, and thirty-two COTS emitters. Intentional Structure Removal (ISR) is performed in two ways: (i) subtracting an ideally generated IEEE 802.11a Wi-Fi preamble from each of the received preambles

in the time domain (a.k.a., the residual representation or error signal [68]) and (ii) dividing the Fourier coefficients of an ideal preamble from each of the received preambles in the frequency domain (a.k.a., the response representation). The authors train an LSTM using a data set of signals at SNR values of 9 dB and 30 dB. These SNR values are achieved by adding realizations of scaled, white Gaussian noise to the received preambles. The authors assess SEI performance using signals that have or have not undergone time or frequency domain-based ISR. The highest SEI performance—when identifying eight emitters—is achieved using the received signals' response representation. When the number of emitters increases to sixteen, SEI performance decreases from 58% to 42% in the frequency domain without ISR at an SNR of 9 dB. When the response representation is implemented, SEI performance decreases from 62% to 48%. When identifying thirty-two emitters, SEI performance is 40% without ISR and 46% when using the response representation at an SNR of 9 dB. Interestingly, SEI performance is higher with thirty-two emitters using the residual representation than when identifying sixteen emitters using the received signals (i.e., ISR is not performed). The authors show that removing the received signals' intentional structure allows the DL network to learn each emitter's RFF features without confusion imposed by the intentional signal structure; thus, improving SEI performance as the number of emitters increases.

Technical Gaps—Increasing Number of Emitters

Based on the papers reviewed in this section and the work presented in [154], it is clear that SEI-based security solutions face a scalability problem in that as the number of emitters increases, the SEI performance suffers. This observation is reinforced by the authors of [155], who state that DL-based model accuracy decreases as the number of to-be-identified emitters increases. The question is whether or not this is a problem with DL or one that plagues even traditional machine-learning approaches. The literature has stated that DL performance improves as the amount of data increases [156], but for DL as it is applied to natural language processing, facial recognition, and spectrum management, and not SEI. Therefore, future research is needed to address this challenge, and the solution may not rest with the DL or machine-learning algorithm(s) but with the signals, their representation, or the *uniqueness* and *permanence* of the SEI-exploited features.

*7.2. Cross-Collection SEI*

Cross-collection SEI refers to the case in which emitter-specific features are learned from a single set of collected signals by a machine-learning algorithm and then used to identify the same emitters using signals collected at another time. Typically, cross-collection SEI involves signals collected over multiple sessions spanning hours, days, weeks, or more. It is important to note that cross-collection SEI assumes the same receiver is used for all signal collections. Cross-receiver SEI is designated receiver-agnostic SEI and is addressed in Section 9.2.

The work in [154] analyzes the effect of IQ channel balancing on cross-collection classification performance across ten thousand emitters. This work utilizes Per-Transmission Accuracy (PTA) and PSA. When using PTA, the entire Wi-Fi QPSK signal's IQ samples are input into the network. When PSA is used, each signal is sectioned into portions, and each portion is input into the network. The emitter with the most classifications across all portions is chosen as the classification for that given signal. In the first experiment, a CNN with ten stacked convolutional layers is trained to identify devices from thirteen USRP N210s and seven X310s. When trained on a single day out of ten days, the average day-zero PSA is 99%. When identifying the same emitters by their signals collected on the remaining nine days, the average PSA falls to 5% with a maximum accuracy of 6% without IQ channel equalization. When IQ channel equalization is performed, the average day-zero PSA falls to 99.5% while the average PSA when classifying the remaining days increases slightly to around 6% with a maximum of 12%. When the number of Wi-Fi devices increases to three hundred and fifty, the ResNet-50-ID CNN can classify day-zero data with

an average PTA of 74.5% and a PSA of 44.1%. When classifying signal collection on another day, the PTA and PSA fall to 1.2% and 1.3%, respectively. When utilizing IQ-channel equalization, the PTA and PSA, when classifying on a new day, increase to 17.5% and 25.8%, respectively. The ten-layer CNN has a PTA of 23.2% and PSA of 22.5% when classifying signal collected on days other than day zero, and IQ-channel equalization is employed. The authors conclude that the drop in classification performance—when training using one day's worth of collected signals and testing using another day's collected signals—is due to the non-stationary nature of the wireless channels the Wi-Fi emitters communicate over.

The authors of [104] investigate the effect of traditional wireless channel mitigation techniques on cross-collection SEI. Ten thousand IEEE 802.11a Wi-Fi preambles are collected from thirty-two TP-Link Archer T3U USB Wi-Fi dongles over eight days, with one week between collections (i.e., signals are collected every Friday). The first experiment trains a CNN on a combination of real, imaginary, magnitude, and phase components of the preambles' time and frequency domain representations. $n$-choose-$k$ components are used where $n = 8$ (i.e., four from each representation), and $k$ is incremented from one to eight in steps of one. A total of two hundred and fifty-five representations are trained and tested. When ranked by the highest cross-collection accuracy, the feature combinations containing the frequency components ranked higher than combinations primarily of time domain components. The combination of all four frequency components is ranked nineteenth with an accuracy of 18.56%, and the combination of all time components is ranked one hundred and ninety-eighth with an accuracy of 16.41%. Based on these results, the authors chose to use only the four frequency components for the remainder of the experiments because they represent only 33% of the total data while reducing the cross-collection SEI performance by approximately 0.7%. When using the frequency representation's real, imaginary, magnitude, and phase components, average SEI performance across all collections fell by 0.75% to 18.56%. Meanwhile, the four time-domain components result in an average accuracy of 16.41%. The authors also perform SEI using CNNs with depths of one, two, four, eight, sixteen, and thirty-two stacked convolutional layers using all four frequency components and only day-zero collected signals. The single convolutional layer CNN achieves an average SEI performance of 15% across all collections, and the four convolutional layers CNN achieves the highest average accuracy of 18.57%. Next, $n$-choose-$k$ preambles collected on day zero are selected for training where $n = 10{,}000$ preambles and the values of $k$ are set to 1000, 2000, 5000, and 10,000. From these selected preambles, one, two, four, eight, and sixteen 30 dB AWGN realizations are generated to simulate multiple instances of the wireless channel for network training. When training on one realization of 1000 preambles, the average SEI performance when classifying each emitter's day-zero collected 10,000 preambles is 55%. When training on sixteen noise realizations added to the same set of 1000 preambles, average SEI performance increased to 61.43%. When the number of preambles increases to 2000, average SEI performance increases to 64.77% with only a single noise realization. When training on a single realization and 10,000 preambles, average SEI performance increases to 95%, showing that real-world collected signals allow the network to generalize better than multiple simulated noise realizations. When training on 10,000 preambles per emitter collected on day zero and classifying across all collections, the accuracy is 19.78% with one noise realization and 22.59% when using sixteen noise realizations. Training on sixteen noise realizations also increases the training time by a factor of sixteen, showing that there are diminishing SEI performance gains for a large increase in training time. Channel effect motivation is investigated through the use of STS symbol averaging. When using STS symbol averaging, cross-collection SEI performance is 20.8% when only the averaged symbol is re-inserted into the preamble, 20.48% when the average STS symbol is replicated and re-inserted into the preamble, and 20.32% when only the replicated STS symbol is used, all of which are below the average SEI performance of 21.53% without STS averaging. Next, the authors use a residual representation of the preambles. In this study, the residual is calculated by dividing the first through fifth STS by the sixth through tenth STS then dividing the first LTS by the second LTS and

finally concatenating the two resulting portions. The residual preamble drops average SEI performance from 20.71% to 11.62%. Finally, the number of collection days represented in the training set is increased from one to seven—in increments of one day's worth of signals–to provide the CNN with a greater presentation of the possible SEI feature variation that can occur across an emitter's transmissions over multiple collections. When training on day-zero collected signals, the average SEI performance when classifying all collections is 20.32%. When adding the second day's signals to the training set, the average SEI performance increases to 38%. This trend continues as each subsequent day's collected signals are added to the training set, resulting in an average SEI performance of 94.4% when the first seven collections are used to classify all eight collections. Despite this, day #8's average SEI performance is never more than 30% when five or more days worth of preambles are included in the training set. This is 22% higher than the same collection's classification accuracy of 9% when only day-zero's collections are used to train the network. The authors conclude that current DL-based SEI techniques cannot adequately learn a general representation of an emitter's RFF features to achieve high accuracy across multiple signal collections.

The authors of [157] leverage Adversarial Domain Adaption (ADA) and device rank to improve emitter classification performance between a set of (i) twenty USRP and (ii) ten HackRF One SDRs. The ADA algorithm utilizes transfer learning between a source, day #1, and a target, day #2, data set. Training is conducted on a (i) Feature Extractor $F$, (ii) Domain Discriminator $D$, and (iii) Source Classifier $C$. $F$ highlights the coloration within a specific emitter's signals and passes them to $D$ and $C$. $D$ classifies the signals regarding whether it is the source or target, while $C$ aims to identify the originating emitter correctly. The goal of $F$ is to maximize the accuracy of $C$ and confuse $D$ to highlight domain-invariant features. The parameters of $F$, $D$, and $C$ are updated using back-propagation until a target performance level is met. $F$ is then attached to a $k$NN classifier. $F$'s parameters are frozen, and the $k$NN's parameters are updated to minimize error when classifying the target data set (a.k.a., day #2). Device rank is a method of identifying an emitter using multiple portions of its collected signal. When classifying, each portion is independently assigned a class by the classifier. The emitter that has the most assignments is the overall classification decision. This is the same approach as the PSA method in [154]. The authors also investigate the effect of the window and stride length using the device rank approach and a CNN. The highest day #1 accuracy is achieved with a window length of 288 and a stride of 1. As the stride length across the IQ signal increases, the day #1 accuracy decreases significantly. Day #2 performance for the USRP data set did not change with the stride length. SEI performance increases from 20.4% to 26.23% when the stride length increases from 1 to 576 samples using the HackRF One data set. When the CNN is replaced with ADA, average SEI performance increases from 8.41% to 43.17% when using the USRP data set and from 25.98% to 65.24% when using the HackRF One data set. The authors attribute this performance increase to the robust nature of the ADA algorithm, but the presented results are not the result of blind testing. The CNN-only method is trained using day #1 signal collections, while the ADA is trained using signals collected on day #1 and day #2; thus, resulting in an unfair comparison because the ADA's SEI results correspond to validation testing while the CNN's SEI results correspond to blind testing.

The authors of [158] use Zero-Shot Learning (ZSL) to cluster unknown, unlabelled emitters. ZSL first learns SEI features of a known, trusted emitter set and maps the emitters into clusters. When new data are presented to the classifier, they either fall into a known cluster and are identified as the emitter assigned to that cluster or do not fall within a known cluster and are held out as a new emitter. ZSL allows for new clusters to be generated online. The authors use eight USRP B210 SDRs. The receiver, sampling frequency, and whether CFO is removed are unclear. Training is performed with a CNN, Multi-Layer Perceptron (MLP), and an AutoEncoder (AE). Five emitters are selected and designated as known, while the remaining three are held out as unknown. Once trained, each NN's activation is passed to the clustering algorithm. Once trained using the known emitters' signals, the

clustering algorithm uses the unlabelled, unknown emitters' signals. SEI performance is determined based on the clustering algorithm's ability to sort and accurately assign each unknown emitter to its unique cluster. Neither the NNs nor the clustering algorithm achieves a blind test accuracy above 50%. Average SEI performance is around 33%, which is a guess when classifying the three unknown emitters. The authors conclude that SEI networks cannot uniquely identify previously unseen emitters well.

Technical Gaps—Cross-Collection SEI

Based on the reviewed work, it can be concluded that a solution for cross-collection SEI has not been found. It can be concluded that a large factor contributing to the poor performance when classifying a new set of collected signals is the non-stationary nature of the wireless channel. AWGN noise realizations do not sufficiently model this channel, but the networks generalize better when training on more real-world, collected preambles. Until cross-collection SEI performance improves, the current methods, whether based on traditional or deep learning, are insufficient for securing IoT deployments over multiple collections; thus, throwing the *permanence* of SEI exploited features into question.

## 8. SEI Data Sets

This section provides a summary of signal data sets that have been used to generate SEI results in published papers and that the authors have made available to the SEI research community to aid further advancements within the topic area and serve as benchmarking data sets.

1.  **POWDER Signals Set:** The POWDER signals set is used to evaluate SEI performance in vendor-neutral hardware deployments of 5G and Open Radio Access Networks (ORANs) [134,159]. The new 5G and ORANs paradigm includes emitters transmitting different protocol signals such as 5G, Long-Term Evolution (LTE), and Wi-Fi at different times. The work in [159] evaluates SEI as a PHY layer authentication technique in such networks using over-the-air signals collected by the large-scale POWDER platform. The signals set includes IQ samples collected from four base stations in different geographical areas. Each base station is implemented using an Ettus USRP X310 SDR and is used to transmit standard-compliant IEEE 802.11a Wi-Fi, LTE, and Fifth Generation-New Radio (5G-NR) frames generated using MATLAB®'s Wireless Local Area Network (WLAN), LTE, and 5G toolboxes. A USRP B210 SDR—located at a fixed point—collects the signals transmitted by the four base stations at a sampling frequency of 5 MHz for Wi-Fi and 7.69 MHz for LTE and 5G. For each base station, the receiver is used to collect IQ samples for two independent days. A single-day collection comprises five sets of IQ samples per base station and protocol, each two seconds long. The data are stored in binary files using Signal Metadata Format (SigMF). Each SigMF file consists of a metadata file containing a description of the collected signals and a data file holding the actual collected signals' IQ samples.

2.  **DeepSig RadioML Signals Sets:** These signals sets are used to evaluate the classification performance of emitter signals in [62,160]. The work in [62] studies the effects of symbol rate and channel impairments on RF signals classification performance by (i) simulating the effects of CFO, symbol rate, and multipath as well as (ii) measuring over-the-air classification performance using software emitters. The signals set used in [62] captures twenty-four different digital and analog single-carrier modulation schemes, including On-Off Keying (OOK), 4-ary Amplitude Shift Keying (ASK), 8-ary ASK, Binary Phase Shift Keying (BPSK), Quadrature Phase-Shift Keying (QPSK), 8-ary Phase Shift-Keying (PSK), 16-ary PSK, 32-ary PSK, 16-ary Amplitude and Phase-Shift keying (APSK), 32-ary APSK, 64-ary APSK, 128-ary APSK, 16QAM, 32QAM, 64QAM, 128QAM, 256QAM, Amplitude Modulation-Single Side-Band-Without Carrier (AM-SSB-WC), AM-SSB with Suppressed Carrier (AM-SSB-SC), AM-Double Side-Band (DSB)-WC, AM-DSB-SC, Frequency Modulation (FM), Gaussian Minimum-Shift Keying (GMSK), and Offset Quadrature Phase-Shift Keying (OQPSK). The resulting mod-

ulated symbols are shaped using a root-raised cosine pulse shaping filter. Channel parameters such as Rayleigh fading delay spread are randomly initialized before each transmission to simulate a time-varying wireless channel. The signals are transmitted and collected using USRP B210 SDRs in an indoor channel on the 900 MHz Industrial, Scientific, and Medical (ISM) band for the over-the-air portion of the signals set. Each captured signal in this data set is composed of 1024 samples. The signals of two million samples are encoded using the hdf5 file format.

Another DeepSig signal set was generated by the authors of [160]. The authors of [160] investigate the feasibility of applying machine learning to the signal processing domain. The authors use the GNU Radio platform to generate a synthetic collection of signals with varying SNR and eleven types of analog and digital modulation, including 8-ary PSK, AM-DSB, AM-SSB, BPSK, Continuous-Phase Frequency-Shift Keying (CPFSK), Gaussian Frequency Shift Keying (GFSK), PAM4, 16QAM, 64QAM, QPSK, and Wide-Band FM (WBFM). For the analog and digital portion of the signal set, the authors use a continuous data source from acoustic voice speech and Gutenberg's works of Shakespeare in ASCII, respectively. The data are organized in a multidimensional float32 vector with a size of

$$N_s \times N_c \times D_1 \times D_2, \tag{2}$$

where $N_s$ refers to the number of signals. $N_c$ is set to one, $D_1 = 2$ refers to the I and Q channels, and $D_2 = 128$ is the number of samples in each signal. The four-dimensional data set is stored in cPickle format to facilitate access and integration of machine-learning platforms such as Keras, Theano, and TensorFlow.

3. **ORACLE Signal Set:** This set of signals was collected by the authors of [161] to evaluate their Optimized Radio clAssification through Convolutional neuraL nEtworks (ORACLE) approach. The authors of [161] evaluate the classification (identification) performance of the proposed approach within static and dynamic channels that are simulated using MATLAB® toolboxes. The ORACLE data set includes signals collected from sixteen USRP X310 emitters transmitting IEEE 802.11a Wi-Fi-compliant frames. A stationary USRP B210 SDR collects all of the IEEE 802.11a Wi-Fi frames at a sampling frequency of 5 MHz and a center frequency of 2.45 GHz. More than twenty million signals are collected for each emitter. Each signal is divided into 128 sub-sequences and stored as float64 in binary files.

4. **WiSig Signal Set:** This signal set is generated by the authors of [162]. It includes ten million IEEE 802.11 Wi-Fi signals collected from one hundred and seventy-four COTS Wi-Fi emitters using forty-one USRP receivers over four captures representing four days. The authors of [162] attempt to address degrading SEI performance due to channel variations caused by using different receivers or signals collected over multiple days. The Wi-Fi signals sent by one hundred and seventy-four Wi-Fi nodes to the AP are captured by forty-one USRP receivers, including B210s, X310s, and N210s. To create the raw WiSig data set, four single-day captures were performed and combined to generate a 1.4 terabyte data set. The collected raw signals are prepossessed to extract the first 256 IQ samples from each Wi-Fi frame with and without channel equalization. The authors provide the steps and scripts to preprocess the collected signals and the data set. For convenience, the authors of [162] subdivided the data set into four smaller subsets:

- **ManyTx:** contains fifty signals for each of the one hundred and fifty emitters and the signals collected by eighteen receivers over four days.
- **ManyRx:** contains two hundred signals for each of the ten emitters and the signals collected using thirty-two receivers over four days.
- **ManySig:** contains one thousand signals for each of the six emitters and the signals collected using twelve receivers over four days.

- **SingleDay:** contains eight hundred signals for each of the twenty-eight emitters and the signals collected by ten receivers in a single day.

The WiSig signal set signals are detected using auto-correlation performed using the Wi-Fi preamble's STS portion and re-sampled to a rate of 20 MHz.

*Technical Gaps—Signal Data Sets*

One of the biggest challenges with machine-learning-based research is access to data. The data sets summarized above are some of the largest and the few publicly available for use in SEI research activities. One of the drawbacks to these data sets is that the signals are transmitted by SDR-based emitters, which do not reflect typical emitters employed by IoT devices. Therefore, there remains a need for public data sets of signals collected from COTS emitters used by IoT devices or in IoT deployments.

## 9. Considerations for SEI in IoT Deployments

IoT devices are typically constrained in terms of power, memory, computation, or combinations thereof to keep the cost of the IoT device down, increase the operating lifetime by extending or maximizing battery life, or both. In this section, we summarize SEI works that consider these constraints. A summary of the considerations for each of the reviewed works in this section is provided in Table 2.

**Table 2.** Areas of focus for the literature reviewed in Section 9.1. Computational resources include experiments considering metrics such as memory requirements and training time. Network training considerations include proposed alternative NN architectures aimed at reducing computational requirements. Data considerations include data processing, reduction, and availability.

| Citation | Consideration | | |
|---|---|---|---|
| | **Computational Resources** | **DL Training** | **Data** |
| [163] | ✓ | ✓ | |
| [164] | ✓ | | |
| [165] | | ✓ | |
| [166] | ✓ | ✓ | |
| [167] | ✓ | | ✓ |
| [105] | ✓ | ✓ | ✓ |
| [106] | ✓ | ✓ | ✓ |
| [168] | ✓ | ✓ | |
| [169] | ✓ | ✓ | |

### 9.1. SEI on Resource-Constrained Devices

The authors of [163] present one of the earliest SEI processes designed to reduce or alleviate computation, energy, and communications overheads associated with performing SEI-based security approaches on resource-constrained IoT devices. The authors accomplish this by offloading the SEI task to Cloud and Edge devices or resources. Initial CNN training is performed in the cloud, and once trained, the CNN is mutually re-trained and made sparse through the use of a progressive weight pruning algorithm [170]; thus, the authors focus on reducing computation and energy requirements during the testing or inference stage instead of during CNN training. The re-trained and pruned CNN is then deployed to the Edge to run on a gateway or another IoT device responsible for relaying information. The authors assess the pruned model's computation and energy reduction performance using a Samsung Galaxy S10, an NVIDIA Jetson TX2 Module, and a Xilinx-ZCU104 Field Programmable Gate Array (FPGA). The authors use two data sets

of five hundred IEEE 802.11 b/g/n Wi-Fi emitters with two hundred seventy-three signals per emitter and fifty ADS-B emitters with two hundred seventy-three signals per emitter. The authors perform SEI using a derivative of the ResNet architecture [171] and assess it and its pruned versions in terms of average percent correct performance, pruning rate, and the number of Floating Point Operations Per Second (FLOPS). Using a pruning rate of $5.4\times$ results in a 0.96% drop in average percent correct classification performance—61.4% for the full ResNet versus 60.44% using the pruned ResNet–while reducing the number of FLOPS by 20% when identifying the five hundred Wi-Fi emitters. For the ADS-B data set, a pruning rate of $5.4\times$ results in a 0.25% drop in average percent correct classification performance—88.53% for the full ResNet versus 88.25% using the pruned ResNet—while reducing the number of FLOPS by 19.3%. Regarding the three hardware platforms, the presented approach increases classification speeds by as much as three times on the Samsung Galaxy S10 and eleven and a half times on the FPGA. The authors only present average classification performance results. Hence, it is difficult to determine how evenly distributed the performance is across individual emitters. They do not assess their approach under degrading SNR conditions but do pose techniques to address the latter under future efforts. However, the biggest concern surrounding the approach in [163] is their use of CFO. CFO is estimated and removed before channel equalization but reinserted once equalization is concluded. CFO's presence is a vulnerability that SEI adversaries can exploit. See Section 6 for details. Therefore, it would be interesting to see how the results presented in [163] would change if CFO is not reinserted. Despite this, the work in [163] does provide a viable means of reducing SEI's burden on Edge IoT devices and shows that SEI can be successfully employed on smartphones.

In [164], the authors present an IoT resource allocation approach that leverages SEI-based security. In particular, the authors focus on IoBT deployments, but as previously stated, IoBT is a form of IoT; thus, we only use IoT in our article. The authors' approach aims to improve IoT network Quality of Service (QoS) through the use of SEI and by optimizing network performance. The former is of particular interest here. The authors propose SEI to control user access in lieu of traditional cryptographic approaches because cryptography systems have higher computational requirements, limiting their use in IoT deployments. Specifically, SEI improves IoT network QoS by identifying and removing malicious users/devices that launch Distributed Denial-of-Service (DDoS) attacks to reduce available network resources (a.k.a., power and channel allocations). Such resources are often limited in IoT deployments, and their reduction negatively impacts network performance optimization. The authors optimize network performance by calculating the IoT network's utility. The utility is calculated using many parameters, including but not limited to the number of sensing devices, power consumption of RF circuits, transmit power of the considered devices, data rate(s), and per-device channel allocation. However, the authors do not consider how performing SEI impacts IoT network utility. It may be that performing SEI is "lumped" into another parameter, such as RF circuit power consumption. If it is, it is unclear whether SEI will be performed on individual IoT devices or the proposed centralized server. Either way, a specific parameter or parameters associated with the performance of SEI should be integrated into the IoT utility optimization calculation and contrasted against the use of cryptography—instead of SEI—to highlight the benefit of SEI-based security in IoT deployments. Lastly, the authors do not present any SEI-related results, which—from a purely SEI viewpoint—seems to limit the contributions of the approach. However, this can be easily remedied by integrating SEI into the IoT network utility optimization calculation.

The authors of [165] make a note of the fact that recent advancements in SEI have been made through the use of DL at the cost of large numbers of hyperparameters that are updated via time-consuming backpropagation, along with the fact that DL structures are not scalable, making them computationally expensive; thus, limiting the practicality of DL-based SEI in IoT devices and infrastructure. The authors address these DL-related issues by proposing an SEI process built on a Broad Learning System (BLS) called Adaptive

Broad Learning (ABL). ABL trades the depth of a DNN for width by increasing the number of nodes that comprise the node layer, replaces time-consuming backpropagation with the pseudo-inverse calculated from the node layer to the output layer, and updates only the new nodes instead of all of the nodes when the network needs to be modified. ABL consists of a node layer—that is composed of feature and enhancement nodes—and an output layer. The feature nodes work directly on the signal's raw IQ samples, while the enhancement nodes' inputs are the output of the feature nodes. The authors assess ABL's SEI effectiveness using two publicly available data sets from [172,173]. The authors' results show that ABL's average SEI performance is on par with the best DL approaches using the first data set [172] and slightly above average when using the second data set [173]. Regarding the second data set performance, the authors attribute the poorer performance to the higher sampling rate, which creates feature redundancy that provides an edge to the DNN architectures. The real benefit of ABL is the reduction in training time, which is a fraction of the time needed to train the DNNs. The authors' approach is novel, but a few considerations must be made. First, the authors note that ABL requires massive, labeled data sets for training; thus, limiting its usefulness in which previously unseen emitters are present in the operating environment. In today's increasing and ubiquitous IoT device deployments, the presence of previously unseen emitters seems inevitable. Second, the authors state that redundant information within the signals causes ABL to overfit. Lastly, one must remember that training does not have to be performed on the IoT device but instead can be performed at a central location initially to perform updates; thus, the SEI-performing device would only need the latest trained model. Despite this, ABL is an interesting approach, and further research is warranted due to its novelty within the SEI space.

Similar to the work in [165], the authors of [166] approach SEI using a broad learning network to lower computational load on the end device to address resource constraints associated with IoT integration. However, the work in [166] differs by using signal feature embedding instead of the node expansion approach in [165]. The authors of [166] call their broad learning-based SEI process Signal Feature Embedded Broad Learning Network (SFEBLN). Signal feature embedding intends to approximate the SEI features through the use of a non-linear transformation with the intent of improving SEI performance. Signal features are generated by performing signal processing before and within the broad learning network. Signal convolution, windowed pooling, and signal shifting are performed before the broad learning network. In contrast, internal signal processing consists of calculating the Discrete Fourier Transform (DFT), Discrete Cosine Transform (DCT), and the Short-Time Fourier Transform (STFT). Additionally, the authors perform broad learning-based SEI using a Central Processing Unit (CPU) to show the feasibility of performing SEI without needing a Graphical Processing Unit (GPU). Assessment of SFEBLN is conducted using a set of ADS-B signals and compared with SEI performed using three DL-based approaches, the ABL approach from [165], and two traditional, handcrafted SEI processes. The DL-based SEI processes are a real-valued CNN, complex-valued CNN, and the multi-scale CNN from [174]. Traditional, handcrafted SEI uses random forest and Support Vector Machines (SVM). The authors consider six scenarios in which the SEI processes each identify 10, 20, 30, 50, 100, or 200 individual ADS-B emitters. SFEBLN results in superior average percent correct classification over both handcrafted, real-valued CNN and multi-scale CNN approaches for all six identification scenarios. SFEBLN is also superior to the complex-valued CNN—in terms of average percent correct classification performance—when identifying forty or fewer ADS-B emitters. The true benefit to SFEBLN is its time advantage over the six alternate SEI processes. SFEBLN can be trained in less than 10 s when 100 or fewer ADS-B emitters are represented in the training set and in less than 13 s when 200 emitters are represented. For SNRs of −10, −5, 0, 5, and 10 dB, the average percent correct classification performance of SFEBLN is superior to all alternate SEI processes except for the case of two hundred ADS-B emitters at an SNR of −10 dB. For this exceptional case, the complex-valued CNN proves to be roughly 10% better but at the expense of huge computing overhead (roughly

2000 times). The authors identify a drawback to SFEBLN that it is susceptible to instability in its results due to the random initialization of the single-layer weight and that temperature, multi-threading, and other unspecified factors exacerbate this instability. To address this, the authors assess SFEBLN stability using Monte Carlo simulation while considering impacts on accuracy, training, and testing times. The authors show that SFEBLN stability remains within acceptable limits, but they do not perform the stability assessment under degrading/low SNR conditions; thus, it is difficult to determine if SFEBLN will remain stable as SNR decreases. In addition, the authors of [166] do not address any of the concerns raised by the authors of [165] and highlighted in the previous paragraph.

The authors of [167] design their SEI process using a systems view to suit real-world operations better. The authors consider training data availability, robustness to unknown operational conditions and uncertainty, channel conditions, and computation limitations. The authors address limitations associated with training data availability by acknowledging that data distributions will change between the training and testing phases, using simulated data for training and fine-tuning real-world data, integrating detection of unknown emitters, and using a limited number of training examples per emitter. The robustness to unknown operational conditions and uncertainty limitations are addressed during training by using a large data set and then tuning or adapting the deployed version, a data set comprising multiple, distinct signal types (e.g., Wi-Fi and ZigBee), and a data set containing signals that represent spoofers or other signals in the area of deployment. Channel condition limitations are addressed by including corrupted signals in the training data. In particular, the authors include signals overlapped in time and spectrum with other signals of the same type and SNR with only the amount of overlap changing and other information to improve performance by leveraging other receiver capabilities, such as the direction of arrival. The authors address the final limitation—computation—by lowering the bit precision of the network weights, the network's depth, the number of filters per layer, and network pruning. The work in [167] is not without its limitations. Primarily, the data they used are not publicly available [152] and the DL network used is developed by a private company, BAE Systems Inc., which could limit the SEI research community's access to it.

The authors of [105] present a data reduction approach that leverages entropy to select the most informative portions of a signal's Time–Frequency (TF) representation. The signal's TF representation is a normalized, grayscale image generated from its GT's complex-valued coefficients. A GT image's most informative portions are selected by comparing a portion's (a.k.a., patch) entropy value to the entropy value of the entire image. If a patch's entropy is equal to or greater than the image's entropy value, then that patch is retained for subsequent SEI. If it is lower than the image's entropy value, the patch is discarded. The presented entropy-informed SEI process outperforms SEI processes that use the signal's raw IQ samples and are comparable to those that use the full GT image at SNRs of 15 dB or greater. Compared to the GT image-based SEI process, memory usage and CNN training times are reduced by 93% and 81%, respectively.

The authors of [106] look to lower the SEI burden on IoT devices and the supporting network by investigating DL-based upsampling impacts on SEI performance. In particular, the authors investigate using a CGAN to upsample the signals collected by IoT devices that measured them using a lower sampling rate. Allowing the IoT device to collect the signals at a lower sampling rate aligns with current IoT design practices [175]. The authors use the CGAN to upsample IEEE 802.11a Wi-Fi preambles collected at sampling frequencies of 2.5 MHz, 5 MHz, or 10 MHz to a sampling frequency of 20 MHz and compare results generated from the upsampled signals to those generated using signals sampled at 20 MHz during collection. They compare SEI results generated using signals upsampled using two other conventional interpolation methods: piece-wise Linear Approximation Interpolation (LAI) and Cubic-Spline Interpolation (CuSI). Additionally, the CGAN upsampled signals' SEI results are compared to those generated using a CNN and the organic sampled signals (i.e., the signals are not upsampled before conducting SEI). The greatest improvement in average percent correct classification performance is achieved when the signals collected at

a sampling rate of 5 MHz are upsampled to 20 MHz. The 5 MHz sampled signals result in an average percent correct classification performance between 84% and 95% for SNR values ranging from 9 dB to 30 dB, respectively. Meanwhile, the CGAN upsampled version of the 5 MHz signals results in an average percent correct classification performance between 92% and 98% over the same SNR values range. However, the use of signals collected at a sampling frequency of 20 MHz results in better performance over those upsampled by the CGAN from 5 MHz to 20 MHz, especially at SNR values below 21 dB. Despite this, the work in [106] provides a potential approach for lowering the resource demands (e.g., memory, power, computation) placed on individual IoT devices; however, SEI performance improvements are needed, and the number of devices needs to be increased.

In [176], the authors present an active Distinct Native Attribute (DNA) fingerprinting process capable of identifying legitimate and counterfeit Wireless Highway Addressable Remote Transducer (HART) adapters using sub-Nyquist sampled signals. The work in [176] differs from that in [106] in that the signals are never upsampled or interpolated. It also differs from the other papers cited in this survey in that the emitters under test are stimulated, and DNA fingerprints are generated or learned from the resulting response(s). In other words, the collected responses are not necessarily produced during normal, unstimulated operations and are collected using a wired setup, although assessment is conducted under simulated, degrading SNR conditions. It is also worth noting that the active DNA fingerprinting process in [176] serves a different purpose than passive SEI processes—including those cited in this survey—in that the work in [176] focuses on identifying counterfeit Wireless HART emitters within the pre-deployment portion of their life cycle to ensure or maintain supply chain integrity. In contrast, passive SEI processes primarily focus on securing communications networks during the deployed/operating period of the emitters' life cycles. The authors of [176] perform DNA fingerprinting using a traditional Multiple Discriminant Analysis (MDA) and CNN-based classifier. For MDA-based DNA fingerprinting, the sub-Nyquist signals' time domain representations of magnitude, phase, and frequency are calculated, each is subdivided into eighteen equal-length sub-regions, the statistics of variance, skewness, and kurtosis are calculated for each sub-region, statistics of each sub-region are sequentially concatenated together along with the statistics calculated across the entirety of each time domain representation, and all statistics from each time domain representation are concatenated together to form a DNA fingerprint. For CNN-based DNA fingerprinting, the same time domain representations are calculated for each sub-Nyquist signal. Still, no further processing is conducted, which leaves feature learning and selection to the CNN. The authors of [176] consider both one- and two-dimensional CNN-based DNA fingerprinting. The one-dimensional case uses only the time domain DNA fingerprints. At the same time, the two-dimensional CNN-based DNA fingerprinting uses both the time and frequency domain representations of the sub-Nyquist signals. The authors do include results generated from Nyquist-sampled signals to facilitate comparative assessment. Ultimately, the two-dimensional CNN-based DNA fingerprinting process proves superior in identifying legitimate Wireless HART emitters at an average percent correct classification rate of 91.6% or better at SNRs of −9 dB and higher. This process correctly identifies counterfeit emitters at an average rate of 91.5% or higher at SNRs of −9 dB and higher. These results are achieved at a sub-Nyquist sampling rate of $1/205^{th}$ that of the Nyquist rate. The Wireless HART signals are collected at a sampling rate of 1 GHz, which yields a sub-Nyquist rate of roughly 4.88 MHz. When considering the sub-Nyquist sampling rate and the results presented by the authors of [176], the presented sub-Nyquist DNA fingerprinting process appears to provide a viable method for alleviating or reducing the burden that the current Nyquist and higher sampling rate-based passive SEI processes place on IoT devices and infrastructure.

The authors of [168] present a lightweight SEI process built on the Gated and sliding Local self-attention transFormer (GLFormer). The authors' approach is inspired by the successful use of the Transformer [177]. This self-attention mechanism allows DL architectures to capture interactions and persistent dependencies in sequential data such as time

series data or signals. GLFormer differs from Transformer and many of its derivatives in that it requires fewer parameters, and its computational complexity is linear versus the quadratic computational complexity of Transformer. GLFormer divides the input signals into shorter sequences or patches, embeds the patches into a token sequence via an embedding layer, and extracts SEI features using the combination of a gated attention unit and sliding local self-attention mechanism. The authors collect signals emitted by fifty maritime vessels' Automatic Identification Systems (AIS) and extract the signals' transient and steady-state portions. The authors compare their GLFormer-based SEI process to four and three alternative SEI and Transformer-based approaches. The four SEI processes are Square Integral Bispectrum (SIB) with SVM, Bi-LSTM, a modified version of ResNet [171,178], and InceptionTime [179]. Meanwhile, the conventional Transformer from [177] is used along with the Swin-Transformer [180] and Convolutions to Vision Transformers (CvT) [181] as alternatives to GLFormer. Regarding average percent correct classification performance, the authors' GLFormer-based SEI process proves superior to all alternative approaches with an accuracy of 96.3% when extracting SEI features from the transient portion of the AIS signals. It is second only to Inception-based SEI when using the AIS signals' steady-state portion (90.1% versus 89.4%). However, the GLFormer-based SEI process's real advantage is its computational complexity reduction. The authors measure computational complexity in millions of FLOPS, and GLFormer requires the fewest FLOPS. GLFormer requires thirty-three Mega-FLOPS for transient-based SEI, which is twenty-five Mega-FLOPS lower than the next fewest of the Swin-Transformer-based SEI process. GLFormer requires sixty-six Mega-FLOPS when performing SEI using the AIS signals' stead-state portion, which is 1735 Mega-FLOPS lower than the Inception-based SEI process (highest SEI performance) and fifty Mega-FLOPS lower than the Swin-Transformer-based approach. GLFormer's computational complexity reduction makes it attractive for IoT deployments, especially if re-training or transfer learning can be performed via cloud, Edge, or Fog computing resources. Despite its advantages, the authors do not assess GLFormer-based SEI under degrading noise or channel conditions but state that future work will investigate GLFormer's performance under degrading SNR conditions. Such a study is necessary to ensure GLFormer is a viable SEI process.

In [169], the authors present a Mahalanobis distance and Chi-squared distribution RF fingerprinting approach focused on providing SEI-based authentication within 5G IoT next-generation networks. The authors show that their approach requires lower training times and fewer resources than five other SEI processes while achieving a higher average accuracy. The five alternate SEI processes include traditional and DL-based techniques that include MDA/ML, SVM, $k$NN, LSTM, and a multi-sample CNN. Additionally, the authors test their approach on an open-source, 5G management and orchestration stack using cloud computing. The authors use a simulated signal set—generated using MATLAB®'s Wireless Waveform Generator toolbox—comprising up to 450 emitters with 100 signals per emitter. The simulated SEI features consist of CFO, amplitude mismatch on the IQ components, phase offset on the IQ components, clock skew, and DC offset. SEI performance is assessed using as few as three to as many as seven of these features; however, results for only CFO, amplitude mismatch, and phase offset are provided. The authors' Mahalanobis distance and Chi-squared distribution RF fingerprinting approach achieve the highest average percent correct classification performance of 99.35% with the poorest performance of 95% generated by the MDA/ML-based SEI process. The authors also assess their SEI process under degrading SNR from 30 dB down to 15 dB and as the number of emitters increases from 50 to 450. Overall, the average percent correct classification performance remains consistent as the number of emitters increases but is negatively impacted by lower SNR values. The average percent correct classification performance is between 92% and 94% at an SNR of 15 dB, which is roughly 4% lower than the 20 dB results regardless of the number of emitters. It would have been beneficial to see individual emitter percent correct classification performance because it would have shown cases of confusion between multiple emitters. The authors assert that their approach is intended

to authenticate legitimate emitters and detect illegitimate emitters but do not provide any results supporting the latter claim. This is important because the authors use CFO as an emitter-identifying feature, and CFO is vulnerable to exploitation by adversaries (see Section 6). Thus, further research should investigate the viability of authenticating legitimate and detecting illegitimate emitters when the CFO is not used as an emitter-identifying feature. The authors state that future work will consider alternate channel conditions and signals collected from actual IoT emitters.

Technical Gaps—SEI on Resource-Constrained Devices

The papers reviewed in this section employ various techniques and approaches to reduce SEI's computational and resource requirements to make it a viable IoT security approach. Most focus on reducing the training time and complexity; however, at least the initial SEI training can be performed offline, where training times and computational resource constraints are less of a factor. Such an approach can be advantageous, and trained models can be updated by repeating the offline training as new signals or data become available or using transfer learning. A few of these papers did explore the use of Edge, Fog, and Cloud computing, which is essential as 5G and next-generation networks are deployed, and such computing resources are integrated to facilitate network operation and management. Future SEI research must consider the challenges of transferring and integrating the trained SEI model(s) within IoT devices. For instance, will the SEI model reside on the individual Edge IoT devices or at a central location such as an AP, BS, or a purpose-built device tasked with monitoring a specific portion of the IoT infrastructure? Such considerations will impact how an SEI model is communicated to the employing device, especially in cases where the Edge device lies dormant for long periods and communicates when only necessary to preserve or extend battery life. Communication of an SEI model will add network overhead, which adds complexity. Lastly, the SEI model-employing device(s) will need to store the trained model, which will increase the usage of limited onboard memory, and the weights, biases, or other model values will more than likely be quantized. This will impact the SEI model's accuracy; thus, future SEI work will need to consider the extent of this impact and how to compensate for it.

*9.2. Receiver-Agnostic SEI*

Despite the amount of SEI research conducted over the past twenty-five-plus years, the attention paid to "receiver-agnostic" SEI has been limited to a handful of publications [178,182–186]. This is attributed to the fact that SEI research has primarily focused on investigating or developing novel signal representations, feature generation approaches, feature selection techniques, machine-learning algorithms, communications standards, or a combination thereof; thus, only a single receiver is employed, and its unintentional features have little to no impact on the SEI process because they are consistent across all of its received signals. However, when considering large IoT deployments in which devices change BSs or APs due to mobility or entering, leaving, and re-entering the network, a single receiver is no longer feasible to ensure effective SEI-based security. Such scenarios create the need for an SEI process—built on a single model or trained NN—to be distributed throughout the IoT infrastructure to reduce complexity and simplify development, deployment, and updates in much the same way Tesla® updates the Artificial Intelligence (AI) of its cars [187,188]. This creates a situation in which the receiver collecting signals for the deployed SEI process differs from that used to train it. Since each receiver's RF front end comprises its own components, sub-systems, and systems, each will impart its own set of unintentional features that differ from those of the receiver used to collect the training signal set. This mismatch between receivers' features leads to poor SEI performance even when the only change is the use of another receiver [182]; thus, effective SEI-based IoT security can benefit from a process or processes that train it to learn a set of signal features that are independent of the receiver used in the signal collection. The result is commonly

referred to as "receiver-agnostic" SEI. This section summarizes works that investigate receiver-agnostic SEI.

The earliest receiver-agnostic SEI investigation is presented in [182]. The authors of [182] adopt a calibration-based approach to achieve receiver-agnostic SEI. Calibration is facilitated by training a Residual Neural Network (ResNN) using a set of "golden" receiver-collected signals. The trained ResNN is used to manipulate or change receiver-specific features in another receiver's collected signals to match those present in the golden receiver's collected signals. The authors consider ten receivers that span a range of capabilities from a high-end signal and spectrum analyzer down to mid-range SDRs, which provides a broad assessment of the presented approach to receiver-agnostic SEI. Each receiver is used to collect signals transmitted by twenty-five ZigBee emitters. The authors compare their approach to using an augmented signal set to train the SEI process. This augmented signal set is constructed using signals collected by multiple receivers. The authors' calibration-based approach achieves superior receiver-agnostic SEI performance compared to this simple case. The authors also assess their calibration-based approach when the signals are collected by $[1, 2, 3, \ldots 9]$ receiver(s) and under degrading SNR conditions. The result is improved SEI performance when using multiple receivers. The authors only use the high-end spectrum analyzer as the golden receiver, so it is unclear if similar receiver-agnostic SEI performance can be achieved when the golden receiver is a lower-end—in terms of SWaP-C—receiver. Such an investigation can determine the minimum cost for implementation by IoT device manufacturers or IoT infrastructure/network administrators. Lastly, the authors do not consider the presence of an RFF-mimicking adversary (see Section 6). The RNN's "re-coloring" nature may increase the similarity between an adversary's mimicked signal features and those present in the original/targeted (a.k.a., the one being mimicked) emitter's signals; thus, increasing attack success.

The authors of [183] investigate mitigation of receiver-specific unintentional signal features using a cooperative approach. The authors consider an SEI process tasked with identifying $N_e = 5$ emitters using signals collected by $N_R = 3$ receivers under the assumption that only a single, unknown emitter is operating at the time all receivers are collecting its signals. The authors decompose the received signals using Empirical Mode Decomposition (EMD), Variational Mode Decomposition (VMD), or Intrinsic Time-scale Decomposition (ITD), which is followed by calculation of the skewness and kurtosis of the decomposed signals. SEI is performed using SVM, a Back Propagation (BP) Neural Network (NN), and an LSTM NN. The authors train an SVM for each known receiver (i.e., $N_R$ SVMs) and identify the emitters using a "maximum wins" voting process. In contrast, the single BP-NN and LSTM are trained using the signals collected by all $N_R$ receivers. Thus, the latter two classifiers are trained to learn features that enable receiver-agnostic SEI. The LSTM using skewness and kurtosis calculated from the ITD decomposed signals achieves the highest average accuracy. The authors do not provide individual emitter performance. In addition, the authors simulate the emitters' and receivers' effects on ideal signals. The emitter effects are IQ imbalance, a spurious tone and carrier leakage, and the PA's non-linear distortion. For the receiver, the authors simulate phase noise, quantization noise, and sampling jitter. Although this approach is good for proof-of-concept demonstration, it is of limited practicality in real-world IoT deployments because emitter features have been shown to change from one transmission to another during normal operation [48]. Additionally, the authors did not investigate cases in which the signals collected by one or more receivers are unavailable to the SEI process due to conditions that would stimulate re-transmission, a common occurrence in wireless communications. IoT deployments can form Wireless Ad hoc NETworks (WANETs) and Mobile Ad hoc NETworks (MANETs); thus, the number of emitters may be equal to or less than the number of receivers. How these topologies impact receiver-agnostic SEI remains an open research question.

In [184], the authors present a Separated Batch Normalization-Deep Adversarial Neural Network (SepBN-DANN) for receiver-agnostic SEI. The authors consider the case when the receiver used to collect the training signals is different than the one used to collect

the testing signals; thus, the approach in [184] only considers a two-receiver case. The receivers are not specified but are stated to be of the same manufacturer and model. The two-receiver case is the impetus behind the authors' use of SepBN because the distributions of the receiver-specific features are not identical across their signal sets. Each receiver's collected signals are used as the training set while the other receiver's signals serve as the testing set. In addition to the use of two receivers, the authors collect the signals of twenty unspecified emitters over three days. Although they collect signals over multiple days, it does not appear that the authors perform cross-collection (a.k.a., multi-day) SEI (see Section 7.2). The fact that the authors do not provide emitter specifics makes it impossible to determine if the emitters are of the same manufacturer, model, or some combination of manufacturers and models. Such information would indicate the SEI difficulty level because serial number discrimination (a.k.a., all emitters are of the same manufacturer and model) remains the most challenging SEI case. Despite this, the authors show that their SepBN-DANN approach can achieve average SEI accuracies of 90% or higher for each of the three days, regardless of which receiver's signals are used for training. The average SEI accuracy computed across days and receiver used to collect the training signals is 95.03% for SepBN-DANN versus 90.18% when using only the DANN and 68.22% when using a CNN. The authors do not provide individual emitter accuracy, the specifics of the signal collection setup (e.g., wired connection, wireless, in an anechoic chamber, etc.), or the SNR of the signals and resulting SEI performance. The use of two receivers can initially appear to be a limiting factor, but not if the training receiver is considered the "golden" receiver and the testing receiver the deployed IoT device performing SEI; thus, providing an opportunity for receiver-agnostic SEI in WANET and MANET configured IoT deployments.

The work in [178] achieves receiver-agnostic SEI by compiling a large data set whose contents include the authorized emitters' signals collected by all receivers. A total of ten LoRa nodes are used as authorized emitters and twenty SDRs as receivers. The set of receivers consists of two USRP N210s, two USRP B210s, two USRP B200s, two USRP B210 Minis, two ADALM Pluto SDRs, and nine RTL-SDR receivers; thus, the receivers span a wide range of SWaP-C requirements. The authors' approach to receiver-agnostic SEI uses an adversarial training architecture comprising a feature extractor and two classifiers. One classifier is tasked with authorized emitter identification, and the other with receiver classification. Each signal undergoes CFO correction and normalization to unit energy. Following CFO correction and energy normalization, data augmentation is conducted in accordance with [189]. Data augmentation is applied to the training signals and achieved by passing each of them through a simulated multipath channel with Doppler effects to improve the feature extractor's and both classifiers' robustness to various conditions present within an operating environment. Every training and testing signal is represented using its spectrogram [189]. The authors assess their receiver-agnostic SEI process under various configurations and conditions to include the number of receivers represented in the training signal set, SNR, homogeneous and heterogeneous receiver configurations within and across the training and testing signal sets, and a six-emitter operational wireless network set up within an office environment without Line-of-Sight (LoS) between the emitters and any of the three receivers. The greatest receiver-agnostic SEI success is achieved using a collaborative approach in which the emitter identity predictions of multiple receivers are combined to form a "fused" prediction. The authors note that SEI accuracy increases as fused predictions increase. Despite the encouraging results presented in [178], the authors only present average accuracy results; thus, there is no way to know how well their approach identifies individual emitters. Overall, constructing a large signal set that spans all receivers is not an issue so long as the receivers do not change (e.g., replaced) and are known before training the SEI process. However, this may not be practical in operational IoT infrastructures because every new receiver deployment would necessitate the collection of large signal data sets and computationally expensive re-training. Another observation is that the best receiver-agnostic SEI performance occurs when the training and testing

receivers are of the same manufacturer and model (e.g., only N210s are used) or when the training receivers are of higher SWaP-C than those used for testing. An example of the latter is when training is conducted using signals collected by the N210s and B210s, but the testing signals are collected using the RTL-SDR receivers. Lastly, it is unclear how the approach—presented by the authors of [178]—would fair or be implemented in a WANET or MANET-configured IoT deployment because such configurations face even stricter onboard limitations (e.g., memory, power, computation) and the receivers can and more than likely would change location(s) within a given portion of the network; thus, changing the authorized emitter signals that a given receiver can collect at any point in time. As previously noted, the latter would require collecting large signal data sets and re-training the affected feature extractors and classifiers.

The authors of [185] investigate two approaches for achieving receiver-agnostic SEI. The authors designate these two approaches as Statistical Distance-based Receiver Agnostic (SD-RXA) and GAN-RXA. Both are trained to train a feature extractor that extracts receiver-agnostic features from the signals of a set of emitters regardless of the receiver used to collect them. SD-RXA is built on the assumption that the receiver- and emitter-specific features are uncorrelated due to asymmetry between their features and random receiver-emitter pairing. However, the authors conclude the SD-RXA is difficult to work with due to the statistical distance between receiver and emitter feature distributions being nontrivial and tricky, the challenge of selecting an appropriate distance between two distributions, and most importantly, the feature extractor's effectiveness in achieving receiver-agnostic SEI cannot be evaluated during training. The GAN-inspired GAN-RXA approach overcomes these difficulties and achieves an average percent correct classification performance of 68% when the GAN-RXA feature extractor is trained using forty emitters and twenty-five receivers and tested using ten emitters and one receiver. This is relatively poor when considering the preponderance of SEI works that achieve average percent correct classification performances of 90% and higher. There may be reasons for this performance disparity. First, the emitters and receivers used for training are mutually exclusive to those used for testing (i.e., no emitter or receiver is used for training and testing). This is important because the feature extractor-learned features will be heavily influenced by the features present in the signals of the training emitters. Although the testing emitters can be of the same manufacturer and model (a.k.a., they only differ in serial number), there are still differences between each emitter's signal features. The authors do not investigate the impact of these differences [185]. Second, they use a portion of the publicly available data set provided by the authors of [162], which includes signals collected over multiple days. This is important because SEI performance has been shown to suffer when training and testing are conducted using signals collected at different times (e.g., across multiple days) even when a single receiver is used (see Section 7.2). This makes it difficult to determine if the poor performance is due to the GAN-RXA approach's inability to learn receiver-agnostic SEI features, issues surrounding cross-collection (a.k.a., multi-day) SEI, or a combination of the two. Lastly, the authors do not remove CFO from the signals and are unclear whether signal energy is normalized to unity. Either or both may be biasing SEI performance—positively or negatively—and make the presented approach susceptible to adversary exploitation (see Section 6).

The authors of [186] aim to achieve receiver-agnostic SEI by treating the features of the different receivers as a data augmentation technique to train a simple Siamese model using unsupervised learning [190]. To capture emitter-specific SEI features, the simple Siamese model is optimized using Local Maximum Mean Discrepancy (LMMD) regularization [191]. The authors' receiver-agnostic SEI process achieves a 95% average accuracy at an unspecified SNR. It is important to note that the authors primarily use simulated emitter and receiver SEI features but evaluate their approach using two USRP X310s as the receivers and four USRP N210 emitters. This is a minimal case and not indicative of a realistic IoT deployment comprising tens to hundreds of devices not constructed using high-end (a.k.a., costing more than $3000 to $15,500 per unit) SDRs. It is also unclear which

results correspond to the simulated emitters and receivers versus the SDR-based evaluation. Many of the presented results show three or more receivers, which indicates the simulated emitters and receivers are used. Using simulated SEI features is a valid approach, but pairing them with actual hardware-based results is essential to determining the value of any SEI-focused contribution.

Technical Gaps—Receiver-Agnostic SEI

A truly receiver-agnostic SEI process must accept signals collected from receivers not present during the SEI training process. In addition, these approaches assume the solution rests in developing more sophisticated DL algorithms. Although the interest in DL is well-founded and warranted, it may not be the best approach; thus, future work needs to look into alternative approaches that are not as demanding in terms of computation, memory, time, and resources to make receiver-agnostic SEI better suited for IoT deployments.

## 10. Supplemental Challenges

This section highlights key challenges facing SEI that were not addressed in the previous sections. Some of these challenges must be addressed to move SEI from a proof-of-concept demonstration to a viable, operational security approach capable of protecting IoT deployments.

### 10.1. Quantization of Deep Learning Models

The demand for less computationally complex, low latency, and high privacy DL algorithms—that can be implemented on Edge computing devices such as IoT–is increasing. A primary challenge facing DL model deployment is that they are large and require large-scale computation resources that are not available in IoT devices [192]. Specifically, the DNN memory cost prevents them from being directly placed into deployed IoT devices. One way to facilitate DL model deployment on IoT devices with limited storage resources is to use low-bit quantization to approximate or convert full precision (e.g., 32-bit float) NN weights and biases to low-bit representations such as 8-bit integers. The following research efforts are examples of applying quantization to DL models to facilitate their integration into embedded systems and IoT devices.

- The authors of [193] present a flexible open-source mixed low-precision library referred to as CMix-NN for low-bit quantization of weights and activations into 8-, 4-, and 2-bit integers. The proposed quantization method targets micro-controller units with a few megabytes of memory and without hardware support for floating-point operations. The quantization library can convert convolutional kernels of CNNs to any bit precision in the 8-, 4-, and 2-bits sets. The authors of [193] used the CMix-NN library to compress, deploy, and evaluate the performance of multiple Mobile-net family models on an STM32H7 microcontroller. The CMix-NN library achieves up to an 8% improvement in accuracy compared to the other state-of-the-art quantization and compression solutions for microcontroller units.

- The authors of [194] present effective quantization approaches for Recurrent Neural Network (RNN) implementations that includes LSTM, Gated Recurrent Units (GRU), and Convolutional Long-Short Term Memory (ConvLSTM). The proposed quantization methods are intended for FPGAs and embedded devices such as low-power mobile devices. The authors of [194] evaluated the performance of their quantization approach using the IMDb and moving MNIST data sets.

Recent SEI research focuses on applying DL to effectively extract inherent and discriminating features from the signals' raw IQ samples or their multi-dimensional representations. Although DL has demonstrated success in identifying emitters using learned SEI features, training large DL models is a computationally complex process requiring more resources than those available to IoT devices.

The authors of [107] assess SEI performance in the presence of adversarial replay attacks that is demonstrated using an IEEE 802.11a Wi-Fi network comprising:

- **Access Point (AP)**: Provides traditional AP functionality as well as SEI. The AP is powered by a Raspberry Pi 4 model B.
- **Authorized Users**: each authorized user is a TP-Link AC1300 USB Wi-Fi adapter and a computer running Ubuntu Linux 16.0.
- **Adversary**: The adversary is implemented using an Ettus USRP B210 SDR powered by an NVIDIA Jetson Nano Developer Kit. The adversary actively learns the SEI features of an authorized emitter and then modifies its own signals' SEI features to match those of the selected authorized emitter before transmission to hinder or defeat the AP co-located SEI process.

The adversary uses a GAN to learn and mimic the SEI features of an authorized emitter. Initially, the GAN is trained using backpropagation on an NVIDIA Tesla K40m GPU. The adversary's Edge computing device—given by the Jetson Nano Developer Kit—has limited storage and computational capability; thus, the authors reduced the GAN's trained generator weight and bias values to 12-bit floats via low-bit quantization. This low-bit quantization allowed the authors of [107] to integrate the original model into a limited memory Edge computing device. Despite the contributions of the work in [107], it did not assess quantization's impact on the effectiveness or success of the adversary's SEI mimicry countermeasure/attack.

*10.2. Unlocking the Secrets of SEI*

The inspiration behind SEI is often attributed to human biometrics such as facial recognition, retinal scanning, and fingerprints. The last is the impetus behind SEI's moniker of RF fingerprinting because in SEI the unintentional signal features serve as the "fingerprint" through which a specific emitter is identified. This is based upon the assumption that every emitter's signal features are sufficiently universal, unique, permanent, and collectible. However, this survey summarizes many SEI works that call into question the uniqueness (see Section 6) and permanence (see Section 7.2) of an emitter's unintentional signal features (a.k.a., its fingerprint). Various SEI publications have expressed RF fingerprints similarly to Equation (1). Although Equation (1) models unique and unintentional behaviors manifest in an emitter's transmitted signal, it does not capture the specific mechanism or mechanisms that form them. In other words, Equation (1) assumes a collective approach in which the cumulative effects of an RF front end's components, sub-systems, and systems are "lumped" together into a few parameters, $\Delta A(t)$, $\Delta f$, and $\Delta \phi(t)$. This "lumped" approach not only seems to be an oversimplification of the RF front end but could also be the reason SEI's premise of uniqueness and permanence is being challenged. This is because Equation (1) and similar models may not provide sufficient feature variability for the RF front end's complex architecture. This is very important when considering the security of large IoT device populations in which SEI uniqueness and permanence are essential to uniquely and consistently identifying any number of emitters, which the authors of [155] support by noting that DL-based SEI accuracy drops as the number of emitters being identified increases. In comparison, we know that every human—that has ever lived or will live—has unique and permanent fingerprints (barring damage that destroys or obscures them). Such an assertion is possible because researchers have determined the mechanism that forms them [195]. However, the mechanism or mechanisms responsible for RF fingerprint formation remain unknown and unexplored, but knowing them is essential in determining SEI's viability as an IoT security approach. Thus, further research is needed to unlock the origins of a particular SEI feature or set of features. For instance, what circuits or components within an RF front-end subsystem or system lead to the variability measured within the collected signals? Is that variability sufficient to provide the same level of identification certainty that exists with human fingerprints? The answers to these questions and many others are key to establishing SEI as a viable security solution within an IoT population of millions of devices.

### 10.3. Availability and Format of Large Signal Data Sets

Section 8 summarized four publicly available signal sets that can and have been used for SEI. As previously noted, these signal sets are constructed using signals transmitted by SDRs, which do not reflect the typical emitters one would expect in low-cost ($30 or less) IoT devices and deployments. Therefore, there is a need for the creation, curation, and dissemination of SEI signal sets that are constructed using COTS emitters. These data sets need to span the wide range of IoT-designated communications standards, which include but are not limited to IEEE 802.11 Wi-Fi, LTE, Bluetooth, BLE, ZigBee, LoRa, and SigFox [196], should be collected at the highest sampling frequencies possible, contain the entirety of each transmitted signal, contain channel-only portions to facilitate SNR calculation and other channel-dependent pre-processing and represent tens to hundreds and if possible over 1000 individual emitters from a variety of manufacturers and models while ensuring that serial number SEI can be performed. Of course, this is assuming that the requisite metadata is included to include receiver specifications, operating bandwidth, center frequency or frequencies, collection settings to include connection type: over-the-air, anechoic, or wired connection, antennas used with their specifications, and collection location (e.g., indoor, outdoor). Lastly, data format and encoding should be done using a known standard and with the highest precision feasible to maximize the preservation of as many SEI features as possible.

### 10.4. Standardization of Language

The authors of [54] are the first to attempt to tackle this SEI challenge; however, since the publication of [54], this remains an open challenge/issue. Although the language used within the SEI community is not a technical challenge preventing SEI's transition from a proof-of-concept demonstration to an effective and robust PHY layer-based security solution used within operational IoT deployments, the lack of language commonality can and does hinder clear communication of an SEI work's focus or purpose, how that work relates to the broader community, and outside the community. So, we reiterate the standard language in [54] and encourage researchers to adopt these definitions in their publications.

- **Classification :** (a.k.a., Identification) The process through which emitters are assigned to different classes or categories. Classification is the result of a *one-to-many* comparison between the emitter's signal or its representation and each of the known classes or categories using a measure of similarity (e.g., distance, probability, etc.).
- **Authentication:** (a.k.a., Verification or Validation) The process through which an emitter's identity—typically a digital one such as a MAC address—is authenticated or verified by performing a *one-to-one* comparison between the emitter's signal or its representation and the stored model or representation associated with the identity claimed by the to-be-authenticated emitter.

### 10.5. IoT-Imposed Temperature Considerations

It is important to note that all emitters undergo some form of temperature change—typically an increase—as they turn on and begin transmitting until some steady-state operating temperature is reached. This "warming up" period is short, but SEI features may change. The typical mitigation strategy to alleviate this temperature-dependent feature variation is to allow the to-be-identified emitter to transmit for some set period (e.g., five to ten minutes) before performing signal collection or SEI. However, many IoT devices enter a "sleep" or "shutdown" state to preserve or extend battery life and turn on only to transmit their data before returning to "sleep". This is an essential IoT-imposed operational constraint that SEI has yet to but must contend with because this period of activity may be too short to allow the IoT device's RF front end sufficient time to reach its steady operating temperature conditions. Future IoT-focused SEI investigations must explore signal feature variation impacts related to this IoT-imposed operating condition.

## 11. Conclusions

This survey reviewed publications to identify technical gaps within SEI that are currently hindering or preventing its use as an IoT deployable PLS solution. Technical gap identification is essential as the number of IoT deployments continues to grow, with the total population expected to reach seventy-five billion by 2025. This growth is alarming because it creates an increasing attack surface over which bad actors do and will continue to exploit individual or sets of IoT devices and IoT-connected critical infrastructure. This alarming fact is exacerbated by the knowledge that a majority—70% or more—of IoT devices use weak or no encryption at all due to limited on-board resources such as memory and power, manufacturing costs that are too high to justify the use of encryption, and challenges associated with deploying, implementing, and managing encryption at scale. Due to these security concerns and challenges, SEI has been suggested to secure IoT devices and their corresponding infrastructure. SEI is advantageous because it is a passive technique that has been shown capable of identifying emitters—down to the serial number—using distinct, inherent, and unintentional features imparted upon their emitted signals by the circuits, sub-systems, and systems that comprise their RF front ends. Another justification for an SEI-based IoT security solution is that SEI does not require modification, interrogation, or insertion of additional functionality or capability into the IoT device being identified. This makes SEI ideally suited because it is backward compatible with existing, deployed, and legacy devices without increasing the computational and resource requirements of the end IoT device(s). SEI has been around for almost thirty years, with significant interest focused on it within the past five to eight years, especially with powerful and successful DL algorithms emerging within facial recognition and image and natural language processing communities. Despite the increased attention and the push to deploy SEI as an IoT security solution, it has mainly remained the focus of academic efforts. To change this, we examined SEI works from the perspective of employing SEI as a practical, effective, and usable IoT security approach. In particular, we reviewed existing SEI works through the lens of SEI's integration and use in resource-constrained IoT devices; thus, we considered works that addressed the impact of the wireless environments channel and temperature, the presence of emitters that actively attempt to obscure or modify their emitter-specific features in an attempt to hinder or defeat an SEI process, the performance of SEI as the number of emitters increased or across multiple collections, the existence of publicly available data sets, the resource limitations of the end IoT devices, and the extraction of receiver-agnostic SEI features. Additionally, this survey considered additional challenges that have not received as much attention within the SEI literature but still hinder SEI's deployment as an IoT security solution. This survey differs from previous SEI surveys because an IoT-centric perspective was assumed when analyzing the SEI literature.

**Author Contributions:** Conceptualization, D.R.R.; Survey Structure and Content, D.R.R., J.H.T. and M.K.M.F.; Project administration, D.R.R.; Resources, D.R.R.; Supervision, D.R.R.; Graphic Visualization, J.H.T. and D.R.R.; Writing—original draft, D.R.R., J.H.T. and M.K.M.F.; Writing—review and editing, D.R.R. All authors have read and agreed to the published version of the manuscript.

**Funding:** The work presented in this article is supported in part by the Tennessee Higher Education Commission (THEC) through the Center of Excellence in Applied Computational Science and Engineering (CEACSE).

**Acknowledgments:** The views, analysis, and conclusions presented in this article are those of the authors and should not be interpreted or construed as representing the official policies—expressed or implied—of the Tennessee Higher Education Commission (THEC) or the Center of Excellence in Applied Computational Science and Engineering (CEACSE).

**Conflicts of Interest:** The authors declare no conflict of interest.

## Abbreviations

The following abbreviations are used in this manuscript:

| | |
|---|---|
| 5G | Fifth Generation |
| 5G-NR | Fifth Generation-New Radio |
| ABL | Adaptive Broad Learning |
| ADA | Adversarial Domain Adaption |
| ADLM | Analog Devices Active Learning Module |
| ADS-B | Automatic Dependent Surveillance-Broadcast |
| AE | AutoEncoder |
| AI | Artificial Intelligence |
| AIS | Automatic Identification System |
| AM | Amplitude Modulation |
| AP | Access Point |
| APG | Average Path Gain |
| ASCII | American Standard Code for Information Interchange |
| ASK | Amplitude Shift-Keying |
| AWG | Arbitrary Waveform Generator |
| AWGN | Additive White Gaussian Noise |
| BER | Bit-Error-Rate |
| BLE | Bluetooth Low Energy |
| BLS | Broad Learning System |
| BP | Back Propagation |
| BPSK | Binary Phase Shift-Keying |
| BS | Base Station |
| CAE | Convolutional AutoEncoder |
| CFO | Carrier Frequency Offset |
| ChaRRNets | Channel Robust Representation Networks |
| CGAN | Conditional Generative Adversarial Network |
| CNN | Convolutional Neural Network |
| ConvLSTM | Convolutional Long Short-Term Memory |
| COTS | Commercial-Off-The-Shelf |
| CPFSK | Continuous-Phase Frequency-Shift Keying |
| CPU | Central Processing Unit |
| CSI | Channel State Information |
| CuSI | Cubic-Spline Interpolation |
| CvT | Convolutions to Vision Transformers |
| C&W | Carlini & Wagner |
| DAC | Digital-to-Analog Converter |
| DCFT | Differential Constellation Trace Figure |
| DCT | Discrete Cosine Transform |
| DDoS | Distributed Denial-of-Service |
| DFT | Discrete Fourier Transform |
| DI | Differential Interval |
| DNA | Distinct, Native, Attribute |
| DNN | Deep Neural Network |
| DoLoS | Difference of the Logarithm of the Spectrum |
| DSB | Double Side-Band |
| EMD | Empirical Mode Decomposition |
| FAR | False Accept Rate |
| FGSM | Fast Gradient Sign Method |
| FLOPS | Floating Point Operations Per Second |
| FM | Frequency Modulation |
| FPGA | Field Programmable Gate Array |
| FRR | False Reject Rate |
| GAN | Generative Adversarial Network |
| GAN-RXA | Generative Adversarial Network-based Receiver Agnostic |
| GFSK | Gaussian Frequency-Shift Keying |
| GLFormer | Gated and sliding Local self-attention transFormer |

| GMSK | Gaussian Minimum Shift-Keying |
| GPU | Graphical Processing Unit |
| GRU | Gated Recurrent Unit |
| GT | Gabor Transform |
| HART | Highway Addressable Remote Transducer |
| ICMP | Internet Control Message Protocol |
| IIR | Infinite Impulse Response |
| InfoGANs | Information maximized Generative Adversarial Networks |
| IoT | Internet of Things |
| IoV | Internet of Vehicles |
| IoBT | Internet of Battlefield Things |
| IoMT | Internet of Military Things |
| IIoT | Industrial Internet of Things |
| IQ | In-phase and Quadrature |
| IQI | IQ Imbalance |
| ISM | Industrial, Scientific, and Medical |
| ISR | Intentional Structure Removal |
| ITD | Intrinsic Time-scale Decomposition |
| JCAECNN | Joint CAE and CNN |
| $k$NN | $k$-Nearest Neighbors |
| LAI | Linear Approximation Interpolation |
| LMMD | Local Maximum Mean Discrepancy |
| LO | Local Oscillator |
| LoS | Line-of-Sight |
| LSTM | Long Short-Term Memory |
| LTE | Long-Term Evolution |
| LTS | Long Training Symbol |
| MAC | Media Access Control |
| MANET | Mobile Ad hoc NETwork |
| MDA | Multiple Discriminant Analysis |
| MDA/ML | Multiple Discriminant Analysis/Maximum Likelihood |
| MIMO | Multiple Input Multiple Output |
| MLP | Multi-Layer Perceptron |
| MMSE | Minimum Mean Squared Error |
| N–M | Nelder–Mead |
| NN | Neural Network |
| OFDM | Orthogonal Frequency-Division Multiplexing |
| OOK | On-Off Keying |
| OQPSK | Offset Quadrature Phase-Shift Keying |
| ORACLE | Optimized Radio clAssification through Convolutional neuraL nEtworks |
| ORANs | Open Radio Access Networks |
| OSTBC | Orthogonal Space-Time Block Code |
| PA | Power Amplifier |
| PAM | Pulse Amplitude Modulation |
| PARADIS | Passive RAdiometric Device Identification System |
| PBA | Per Batch Accuracy |
| PCA | Principal Component Analysis |
| PGD | Projected Gradient Descent |
| PHY | Physical |
| PLA | Physical Layer Authentication |
| PLL | Phase-Locked Loop |
| PLS | Physical Layer Security |
| PMF | Probability Mass Function |
| POWDER | Platform for Open Wireless Data-driven Experimental Research |
| PSA | Per Slice Accuracy |
| PSK | Phase Shift-Keying |
| PTA | Per-Transmission Accuracy |
| QAM | Quadrature Amplitude Modulation |

| | |
|---|---|
| QoS | Quality of Service |
| QPSK | Quadrature Phase Shift-Keying |
| RECAP | Radiometric signature Exploitation Countering using Adversarial machine learning-based Protocol |
| ResNN | Residual Neural Network |
| RF | Radio Frequency |
| RFF | Radio Frequency Fingerprint |
| RFFE | Radio Frequency Fingerprint Embedding |
| RF-DNA | Radio Frequency-Distinct, Native, Attributes |
| RNN | Recurrent Neural Network |
| SC | Suppressed Carrier |
| SDR | Software-Defined Radio |
| SD-RXA | Statistical Distance-based Receiver Agnostic |
| SEI | Specific Emitter Identification |
| SepBN-DANN | Separated Batch Normalization-Deep Adversarial Neural Network |
| SFEBLN | Signal Feature Embedded Broad Learning Network |
| SIB | Square Integral Bispectrum |
| SigMF | Signal Metadata Format |
| SNR | Signal-to-Noise Ratio |
| STS | Short Training Symbol |
| STBC | Space-Time Block Code |
| STFT | Short-Time Fourier Transform |
| SSB | Single Side-Band |
| SVM | Support Vector Machines |
| SWaP-C | Size, Weight, and Power-Cost |
| SYNC | Synchronization |
| TeRFF | Temperature-aware Radio Frequency Fingerprinting |
| TF | Time-Frequency |
| USRP | Universal Software Radio Peripheral |
| VCP | Voltage Controlled Oscillator |
| VMD | Variational Mode Decomposition |
| WANET | Wireless Ad hoc NETwork |
| WBFM | Wide-Band Frequency Modulation |
| WC | Without Carrier |
| Wi-Fi | Wireless-Fidelity |
| WLAN | Wireless Local Area Network |
| ZSL | Zero-Shot Learning |

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
