# Peer review of "Considerations, Advances, and Challenges Associated with the Use of Specific Emitter Identification in the Security of Internet of Things Deployments: A Survey"

_information, doi:10.3390/info14090479_

Round 1

Reviewer 1 Report

I am very thankful for reviewing this manuscript and it was my pleasure to assist you with my suggestions. I have read for reviewing assessment and my suggestions have been given in the following points: Why was SEI used or considered by the organisation as part of its IoT security strategy? What are the main advantages of utilising SEI to secure IoT? How does the SEI improve the security posture of IoT networks and devices? What were the main difficulties you faced when applying SEI in IoT deployments? What new developments or breakthroughs in SEI technology have improved IoT security recently? Do the use of SEI in IoT deployments raise any privacy concerns for you? How can you use SEI for security while taking data privacy concerns into account? How do you think SEI will change to handle the difficulties of protecting IoT deployments? Novelty is missing in abstract section. Add research questions, research objectives and research gaps. There is a significant lack of relevant literature references so need some more literature; in order to motivate the researchers in the subject. Add societal implications also. Conclusions section is missing. Mention some application based work that can be applied to today’s rapidly changing environment. Need to improve the English language and general communication.

see the comments.

Author Response

The authors want to thank you for your review, comments, and time that assist in making our manuscript better.

Reviewer 2 Report

This is a very comprehensive review of literature around SEI’s within the context of IoT security. The manuscript is very well written, and entails a great collection of pertinent literature in the field. I have summarised my comments in the following bullet points and I hope the authors find them useful: 

  1. Abstract: The abstract is generally well written, however, in my view, it is quite wordy. To maintain conciseness, using only the second half and a few sentences from the first half of the abstract should suffice. Although I am unaware of the Publisher’s word limits for the abstract, keeping it succinct will be beneficial in serving its purpose.
  2. Introduction: This is also very well written. I do not have a specific comment for this section, except for one related to line 30 i.e., “By the year 2025, the number of IoT deployments is projected to reach seventy-five billion”; is this the number of devices? Deployment may denote implementation of sensory networks comprised of hundreds of devices. It is also worth noting that I found Table 1. very useful, as it substantiates the need for a study such as the one under review.
  3. Sections 2-5 provides a detailed review of literature relevant to each section. What is unclear however is that how the authors identified/collected this literature? I appreciate that this is not necessarily a systematic review and hence it does not require a detailed protocol and reporting guideline, nonetheless, I would like to know: how did you ensure that key literature are not omitted? This information could be provided in a sub-section prior to Section 2, outlining the approach adopted to identify, select and review existing literature. The academic and non-academic repositories searched, and the keywords used for searching for studies could also be outlined in here. Since, this is not an SR, inclusion and exclusion criteria may not necessarily be required.
  4. Section 6 is also a useful addition, and there is a fine justification for its inclusion i.e., reviewing studies that could not be framed as part of the main categorisation.
  5. Section 7 focuses on relevant security considerations. In my opinion, the quality of the review would have been improved if a Table was presented summarising these considerations whilst citing relevant literature. This comment would also be relevant to Sections 2-5, however the need for a Table seems more pressing for Section 7.
  6. Conclusions section is written well .

 All in all, this is a very comprehensive review/survey of the SEI for IoT deployment. The review is articulated well, and the structured appropriately. As indicated above, one or two Tables summarising the key findings from the narratives would make the study a great resource for future related research. 

The work is articulated excellently. Nonetheless, a round of proofreading could help identify any overlooked typographical mistakes.

Author Response

(The authors gave the same response as above.)

Round 2

Reviewer 1 Report

I have read this paper, I would like to point out that some comments and supporting statements should be added to improve the quality. My suggestions are the following: How do you decide SEI publications from the perspective of its use as a practical, effective, and usable IoT security mechanism? Can you explain and highlight other security mechanism? Elaborate the concept of SEI in terms of advantageous. Add more literature on deep learning. The work seems to be very general. It needs to be more specific to add value to the literature. The variables you are studying have been excessively tackled in the literature so make sure to cover that in your work. It would be better if you added some attractive new knowledge, what you reached is very well known. Research questions and research objectives, and research gaps are missing. Add theoretical, managerial and societal implications. Elaborate conclusions with limitations and future scope. 

see the comments.

Author Response

The authors want to thank you for your review, comments, and time that assist in making our manuscript better. The attached PDF contains our response to each of your comments and a highlighted version of the manuscript capturing specific portions that relate to our responses. 
